# Single cell transcriptional evolution of myeloid leukemia of Down syndrome

Mi K. Trinh [1], Konstantin Schuschel[2,3], Hasan Issa[2,3], Rebecca Thomas[4], Conor Parks [1], Agnes Oszlanczi [1], Toochi Ogbonnah[1], Di Zhou[1], Lira Mamanova [1], Elena Prigmore [1], Emilia R. Robertson[4], Angus Hodder[1,4], Anna Wenger [1], Nathaniel D. Anderson [1], Holly J. Whitfield [1], Taryn D. Treger[1,5,6], José Gonçalves-Dias[2,3], Karin Straathof [4,7,8], David O'Connor [4,9], Matthew D. Young[1], Laura Jardine [1,10], Stuart Adams [4], Jan-Henning Klusmann[2,3,11] ✉, Jack Bartram [4,9] ✉ & Sam Behjati [1,4,5,6] ✉

Children with Down syndrome have a 150-fold increased risk of developing myeloid leukaemia (ML-DS). Unusually for a childhood leukaemia, ML-DS arises from a preleukaemic state, termed transient abnormal myelopoiesis (TAM), via a conserved sequence of mutations. Here, we examine the relationship between the genetic and transcriptional evolution of ML-DS from natural variation; a rich collection of primary patient samples and foetal tissues with a range of constitutional karyotypes. We distil transcriptional consequences of each genetic step in ML-DS evolution, utilising single-cell mRNA sequencing, complemented by phylogenetic analyses in progressive disease. We find that transcriptional changes induced by the TAM-defining *GATA1* mutations are retained in, and account for most of the ML-DS transcriptome. The *GATA1* transcriptome pervades all stages of ML-DS, including progressive disease that had undergone genetic evolution. Our approach delineates the transcriptional evolution of ML-DS and provides an analytical blueprint for distiling consequences of mutations within their pathophysiological context.

Children with constitutional trisomy of chromosome 21 (i.e., Down syndrome) have a 150-fold increased risk of developing myeloid leukaemia[1–3]. Myeloid leukaemia of Down syndrome (ML-DS) evolves from a preleukaemic state, termed transient abnormal myelopoiesis (TAM), which may appear and resolve unnoticed in the postnatal period. Approximately 25% of children with Down syndrome have molecular evidence of TAM at birth[4]. Morphologically, the leukaemic cells (blasts) of ML-DS and TAM are largely similar[5], exhibiting megakaryoblastic differentiation and characteristic immunophenotypic features (e.g., CD41, CD61 and lineage-atypical

features e.g., CD7, CD56[6]). There are no distinctive diagnostic features that separate TAM from ML-DS. The key difference lies in the clinical trajectory of these diseases. ML-DS is invariably fatal if left untreated; it is cancer. By contrast, TAM is a self-limiting clonal expansion. It only requires treatment when resolution needs to be hastened because of significant physiological perturbations[7,8]. In clinical practice, age cut-offs are applied to distinguish between TAM and ML-DS, with the former being confined to the first few months of life (6 months in UK practice (http://www.cclg.org.uk/, Tunstall et al., 2018[8])).

[1]Wellcome Sanger Institute, Hinxton, UK. [2]Department of Pediatrics, Goethe University Frankfurt, Frankfurt, Germany. [3]Frankfurt Cancer Institute, Frankfurt am Main, Germany. [4]Great Ormond Street Hospital for Children NHS Foundation Trust, London, UK. [5]Department of Paediatrics, University of Cambridge, Cambridge, UK. [6]Cambridge University Hospitals NHS Foundation Trust, Cambridge, UK. [7]UCL Cancer Institute, London, UK. [8]Great Ormond Street Biomedical Research Centre, London, UK. [9]UCL Great Ormond Street Institute of Child Health, London, UK. [10]Biosciences Institute, Newcastle University, Newcastle upon Tyne, UK. [11]German Cancer Consortium (DKTK), Partner Site Frankfurt/Mainz and German Cancer Research Center (DKFZ), Heidelberg, Germany. ✉e-mail: Klusmann@leukemia-research.de; Jack.Bartram@gosh.nhs.uk; sb31@sanger.ac.uk

ML-DS has been extensively studied. As a stepwise model of leukemogenesis, it may provide insights generalisable to other leukaemia subtypes where a precursor phase cannot be examined. In particular, the genetic basis of TAM and ML-DS has been resolved in detail (Fig. 1A). The first step towards transformation is constitutional trisomy 21. It biases foetal hematopoiesis towards erythroid and megakaryocytic lineages at the cost of the B cell lineage[9–12]. This is followed by the acquisition of oncogenic mutations in the GATA1 gene, which underpin the development of TAM[13–15]. GATA1 is a transcription factor that plays a critical role in regulating erythroid and megakaryocytic fate, via upregulation of genes such as FOG1 and KLF1 and downregulation of genes including KIT, MYC and GATA2[16,17]. Mutations in TAM yield a truncated GATA1 protein, which permits some differentiation but fails to adequately suppress pro-proliferative features[18,19]. In addition to GATA1 mutations, ML-DS harbours cancer causing ("driver") variants in 75–93% of cases[1,20], commonly affecting genes encoding JAK kinases, the cohesin complex, or epigenetic regulators. Between 7 and 25% of ML-DS cases do not acquire further discernible driver events, yet clinically represent a malignant disease rather than a self-resolving clonal expansion of blood. Extensive phenotypic, genetic and detailed modelling work notwithstanding, the effects of each leukaemogenic event in the evolution of ML-DS remains incompletely understood[15,21].

The advent of single-cell transcriptomics provides an opportunity to directly distil cellular consequences of mutations from natural variation. It delivers with every cell an independent readout of an individual's disease, whilst studying the same condition across patients enables the identification of overarching signals. The key advantage of this approach is that it examines variants within their physiological context, rather than in the engineered environment of model systems. A central limitation is the availability of adequate patient samples (fresh material or viably frozen cells), especially in rare diseases such as TAM and ML-DS.

We set out to distil the cellular difference between TAM and ML-DS from natural variation by reconstructing the transcriptional evolution of ML-DS via trisomy 21 and TAM. Harnessing tissue banking efforts of the UK's largest children's cancer unit and a German cohort from an international ML-DS study, we are able to examine the entire spectrum of TAM and ML-DS directly from patient samples, including rare cases of refractory and recurrent disease, in the context of human foetal tissues and other relevant leukaemias.

## Results

### Study cohort overview
We assembled a cohort of primary patient samples from the UK and Germany encompassing TAM and ML-DS, including rare cases of recurrent TAM, recurrent ML-DS, and refractory ML-DS (Table 1, Supplementary Data 1). All children were younger than 5 years at ML-DS diagnosis, and all cases showed megakaryoblastic differentiation on flow cytometry. In addition, we studied other leukaemias to disentangle the effects of mutations, cell lineage, and phenotype on transcription: an exceptional rare case of diploid myelodysplastic syndrome (MDS) harbouring a GATA1 mutation, as well as a case of MDS without GATA1 mutation; acute megakaryoblastic leukaemia (AMKL) in disomic children (n = 2); other childhood myeloid leukaemias (AML, n = 5) to capture myeloid cancer phenotypes; B cell lymphoblastic leukaemias (B-ALL) arising in children with (n = 8) or without (n = 2) Down syndrome, to assess the effects of trisomy 21 in a different haematopoietic line; and a solid haematological B cell neoplasm arising in a child with Down syndrome (central nervous system lymphoproliferative disorder; LPD). To study the effects of trisomy 21 on foetal haematopoiesis, we examined foetal livers (thought to be the origin of preleukaemic cells in Down syndrome[22]) from diploid donors and donors with variant constitutional karyotypes, including trisomy 21 (T21), trisomy 18 (T18), trisomy 22 (T22), monosomy X (MX) and

whole genome trisomy or triploidy (Supplementary Data 1). All samples (fresh or viably frozen) were enriched for live cells prior to single-cell mRNA sequencing (scRNA-seq) using the Chromium 10X platform. We processed the scRNA-seq data in a standardised way, including removal of ambient mRNAs and doublets, to derive high-quality count tables of transcripts for 653,394 foetal liver cells, and 525,650 patient cells: 196,090 normal and 329,560 neoplastic cells. In addition, we performed whole-genome sequencing where possible to enable integrated analyses of transcriptional and genetic evolution, as detailed in individual sections. A detailed account of the study cohort, including assays and data volume, is provided in Supplementary Data 1.

### Effects of karyotypic variants on foetal hepatic haematopoiesis
To establish the contribution of trisomy 21 to the ML-DS transcriptome, we compared haematopoietic cells from trisomic 21 fetuses with disomic foetal liver cells. We expanded our analyses to liver haematopoietic cells from fetuses with a range of karyotypic variants—trisomy 18, trisomy 22, monosomy X and triploidy (Fig. 1B, Supplementary Data 1)—to establish whether the changes induced by trisomy 21 are specific to Down syndrome or represent a generic perturbation of haematopoiesis caused by numerical variation of the human karyotype. Overall, our foetal liver scRNA-seq dataset captured 31 different cell types across both haematopoietic and stromal compartments, with representation of all karyotypes (Fig. 1C, Supplementary Fig. 1). In keeping with previous work[9,10,12], we found that in trisomy 21, the megakaryocyte-erythroid-mast cell compartment significantly expanded in the foetal liver at the expense of the B cell lineage (Fig. 1D). Other non-disomic karyotypes also perturbed the composition of the haematopoietic compartment, albeit subtly and in different directions. An exception was the triploid foetal liver, which showed triplication of chromosome 21 and exhibited perturbations similar to those observed in trisomy 21. This suggests that the expansion of the megakaryocyte-erythroid-mast cell compartment observed in the triploid foetal liver is predominantly driven by trisomy 21, consistent with a view that the addition of chromosome 21 directly underpins perturbed haematopoiesis. To further evaluate the transcriptomic impact of trisomy 21 relative to other karyotypes, we compared gene expression between non-disomic cells and their diploid counterparts. Across all abnormal karyotypes and cell types, we observed genome-wide transcriptional perturbation, with most genes on the trisomic chromosomes exhibiting increased expression (Supplementary Fig. 2A, Supplementary Data 2). Trisomy 21 had a less pronounced impact on global transcription compared to other single trisomies (Fig. 1E). Nonetheless, the transcriptional changes in trisomy 21 cells were largely karyotype-specific (Fig. 1F). Overall, these analyses demonstrate that the foetal haematopoietic compartment is sensitive to numerical variations in the constitutional chromosome complement, with specific effects imparted by trisomy 21.

### Stepwise transcriptional progression towards ML-DS
We set out to characterise transcriptional changes at each step along the development of ML-DS, building on single transcriptomes we generated from nine children with TAM and 11 with ML-DS (Fig. 2A, Table 1, Supplementary Data 1). In the first instance, we performed Louvain clustering of all cells together and generated a Uniform Manifold Approximation and Projection (UMAP) representation of 375,548 high quality cells across the dataset (Fig. 2B, Supplementary Fig. 3). We were able to unambiguously identify clusters of neoplastic TAM and ML-DS by interrogating the nucleotide sequences derived from scRNA-seq for the presence of GATA1 mutations (Fig. 2C, Supplementary Fig. 4A). As is usually seen in UMAPs of single neoplastic cell transcriptomes, neoplastic cells mostly formed patient specific clusters (Supplementary Fig. 3B), further indicating that TAM blasts are transcriptionally transformed. Of note, a small population of GATA1-mutant neoplastic cells from the refractory sample of patient L038

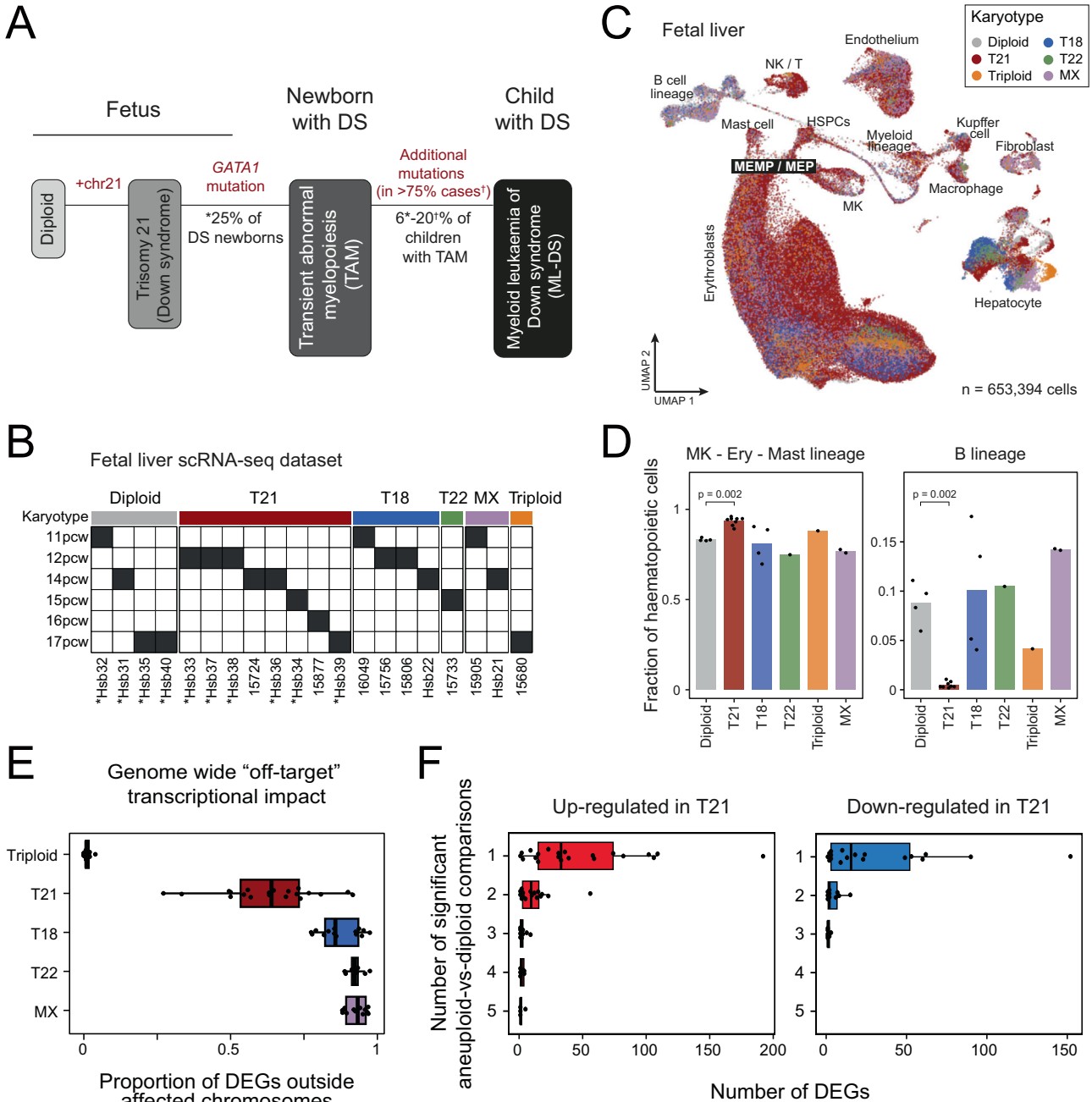

**Fig. 1 | Transcriptional effects of karyotypic variants on foetal hepatic hae-matopoiesis. A** ML-DS stepwise pathogenesis. Frequency of *GATA1* mutations in neonates with Down syndrome (DS) was from (*) Elliott et al., 2025[4]; and number of ML-DS cases with known additional driver mutations was from (†) Labuhn et al., 2019[1] and Sato et al., 2024[20]. **B** Foetal liver samples from fetuses with different karyotypes. Asterisks (*) indicate samples where cells were sorted into CD45+ and CD45- fractions before sequencing. PCW post conception week. **C** Uniform mani-fold approximation and projection (UMAP) visualisation of the foetal liver scRNA-seq dataset, with cells (dots) coloured by karyotype. HSPCs haematopoietic stem and progenitor cells; MEMP / MEP – megakaryocyte-erythroid-mast cell progenitor / megakaryocyte-erythroid progenitor; MK – megakaryocyte; NK / T – natural killer cell / T cell. **D** Bar plot showing the mean proportion of megakaryocyte-erythroid-mast cell compartment and B lineage among all haematopoietic cells across donors for each karyotype. Dots represent individual foetal samples within the karyotype

group. Trisomy 21 samples (*n* = 8) showed significantly increased megakaryocyte-erythroid-mast cell proportion and reduced B lineage representation compared to diploid samples (*n* = 4; *p*-value = 0.002 for both, one-tailed Wilcoxon rank-sum test). **E** Box plots showing the genome-wide "off-target" transcriptional impact of different karyotypes across cell types, measured as the proportion of differentially expressed genes (DEGs) located outside the affected chromosomes (x-axis). Each dot represents a cell type with sufficient cell number to be compared against diploid cells, using a pseudo-bulk differential expression analysis approach (see Methods). Boxes indicate first and third quartiles, central line denotes the median, and whiskers extend to 1.5× IQR. **F** Number of differentially expressed genes identified in each trisomy 21 cell type (dots) when compared to diploid counter-parts. DEGs are further grouped by the number of other abnormal karyotypes in which they also appeared as DEGs (indicated by the number of comparisons – x axis). Boxplots are defined as in (**E**). Source data are provided as a Source Data file.

**Table 1 | Overview of study cohort**

| | | Conditions | Focus of analysis | Individuals | Samples |
|---|---|---|---|---|---|
| Single-cell RNA sequencing dataset | **Foetal liver** | Diploid cells | Trisomy 21-specific effects on foetal hepatic haematopoiesis | 4 | 4 |
| | | Down syndrome | | 8 | 8 |
| | | Other aberrant karyotypes | | 8 | 8 |
| | **TAM & ML-DS** | Conventional TAM | Effects of *GATA1* mutation | 8 | 8 |
| | | Recurrent TAM | Unusual clinical phenotype | 1 | 2 |
| | | ML-DS | Difference to TAM | 9 | 17 |
| | | Relapse ML-DS | Unusual clinical phenotype | 1 | 4 |
| | | Refractory ML-DS | Unusual clinical phenotype | 1 | 2 |
| | **Diploid myeloid neoplasms** | AMKL | Differences between TAM/ML-DS and its morphological mimic | 2 | 2 |
| | | Myelodysplastic syndrome with *GATA1* mutation | Effects of *GATA1* mutation without trisomy 21 | 1 | 1 |
| | | Myelodysplastic syndrome without *GATA1* mutation | Specificity of *GATA1*-mutation induced transcriptome | 1 | 1 |
| | | Other myeloid leukaemia | Specificity of ML-DS transcriptome | 5 | 5 |
| | **Lymphoblastic neoplasms** | B-ALL leukaemia in Down syndrome | Specificity of findings to myeloid lineage and Down syndrome | 2 | 2 |
| | | CNS lymphoproliferative disease in Down syndrome | | 1 | 1 |
| | | B-ALL without Down syndrome | | 8 | 9 |
| Validation dataset bulk RNA sequencing | | Normal haematopoietic stem and progenitor cells | Validating the observations derived from single-cell mRNA sequencing data | 15 | 60 |
| | | TAM | | 15 | 16 |
| | | ML-DS | | 13 | 13 |
| | | AMKL | | 10 | 10 |
| | | *KMT2A*-rearranged leukaemia (mixed-lineage leukaemia / MLL) | | 26 | 28 |

Summary of study datasets showing cohort composition, analytical focus, and numbers of individuals and samples analysed.

clustered with normal erythroblasts. As these blasts share the same *GATA1* mutation as the main malignant population, these cells likely represent a neoplastic subclone that has adopted an erythroid-like transcriptional state. This may indicate a more lineage-committed phenotype, potentially enabling a subclone to evade chemotherapy, perhaps through a less proliferative state, although further investigation is required to substantiate this hypothesis.

Next, we performed a series of differential gene expression analyses to quantitatively determine the transcriptional changes imparted by trisomy 21 followed by *GATA1* mutations along the progression towards ML-DS (see Methods). We first defined the ML-DS transcriptome as the difference between ML-DS cancer cells and diploid foetal liver megakaryocyte-erythroid-mast progenitor cells (MEMP/MEP). These differentially expressed genes represent the transcriptional changes between a completely normal diploid state and leukaemia. We found that only 37 genes (23 up-, 14 down-regulated), accounting for 12% of the ML-DS transcriptome, were significantly perturbed in the same directions in trisomy 21 foetal hepatic MEMP/MEP cells when compared to the diploid state (Fig. 2D, Supplementary Data 3). Among those upregulated were five genes on chromosome 21, along with several other genes previously implicated in leukaemogenesis through promoting cell proliferation (*ITGB7*[23], *ADAM8*[24,25]), as well as genes known to be highly expressed in mature megakaryocytes (*ITGA2B*[26]), mast cells (*FCER1A*[27], *LTC4S*[28]) and erythroblasts (*CA1, CA2*[29]). Additionally, some haematopoietic progenitor markers, such as *HOXA9*[30] and *DACH1*[31], were downregulated, suggesting a more differentiated transcriptional state in ML-DS blasts.

We then assessed the contribution of *GATA1* mutation towards the leukaemic transcriptome. This is defined as the transcriptional changes between trisomy 21 bone marrow MEP and ML-DS blasts, which

have already occurred in TAM upon *GATA1* mutation acquisition. The analysis revealed that the majority (>80%) of differentially expressed genes between ML-DS blasts and trisomy 21 MEP are in fact already significantly perturbed in TAM blasts (Fig. 2E, Supplementary Data 4). Eight genes on chromosome 21 were found to be significantly upregulated in cells with *GATA1* mutation, including those previously described to be involved in ML-DS (*MIR99AHG*[32], host gene of miR-125b-2) or general cancer development (*PRMT2*[33], *AGPAT3*[34]). Moreover, *GATA1* mutation resulted in up-regulation of its downstream targets (*IRX1*[35]), along with many genes normally expressed in definitive megakaryocytes, mast cells (*MPL, NR4A1*) or myeloid lineage (*ZBTB16*), but down-regulation of erythroid-specific genes (Supplementary Fig. 5A). This is consistent with data on functional consequences of *GATA1* mutation from in vitro assays of patient-derived iPSCs[36] and mouse models[32,37,38]. All three lines of evidence demonstrate that *GATA1* impairs erythropoiesis while promoting a myelomegakaryocytic bias. Many genes involved in stem cell maintenance and leukaemogenesis were also up-regulated (*IGF2BP1, EPHA3, MLF1, IL1RAP, TGFB1, TGFBR2*).

Nonetheless, trisomy 21 and *GATA1* mutations together did not fully recapitulate the gene expression profile, and thus the phenotype, of overt ML-DS. To distil the remaining differences required for the final step of leukaemogenesis, we compared the transcriptional profiles of diagnostic (treatment-naive) ML-DS cancer cells with those of non-recurrent TAM. We identified 198 genes differentially expressed across multiple ML-DS and TAM individuals, many of which have known roles in myeloid leukaemia, such as *ARID3A*[39] – the downstream target of miR-125b (Fig. 2F, Supplementary Data 5). However, we observed large patient-specific transcriptional changes when comparing blasts from each ML-DS individual against TAM (Supplementary

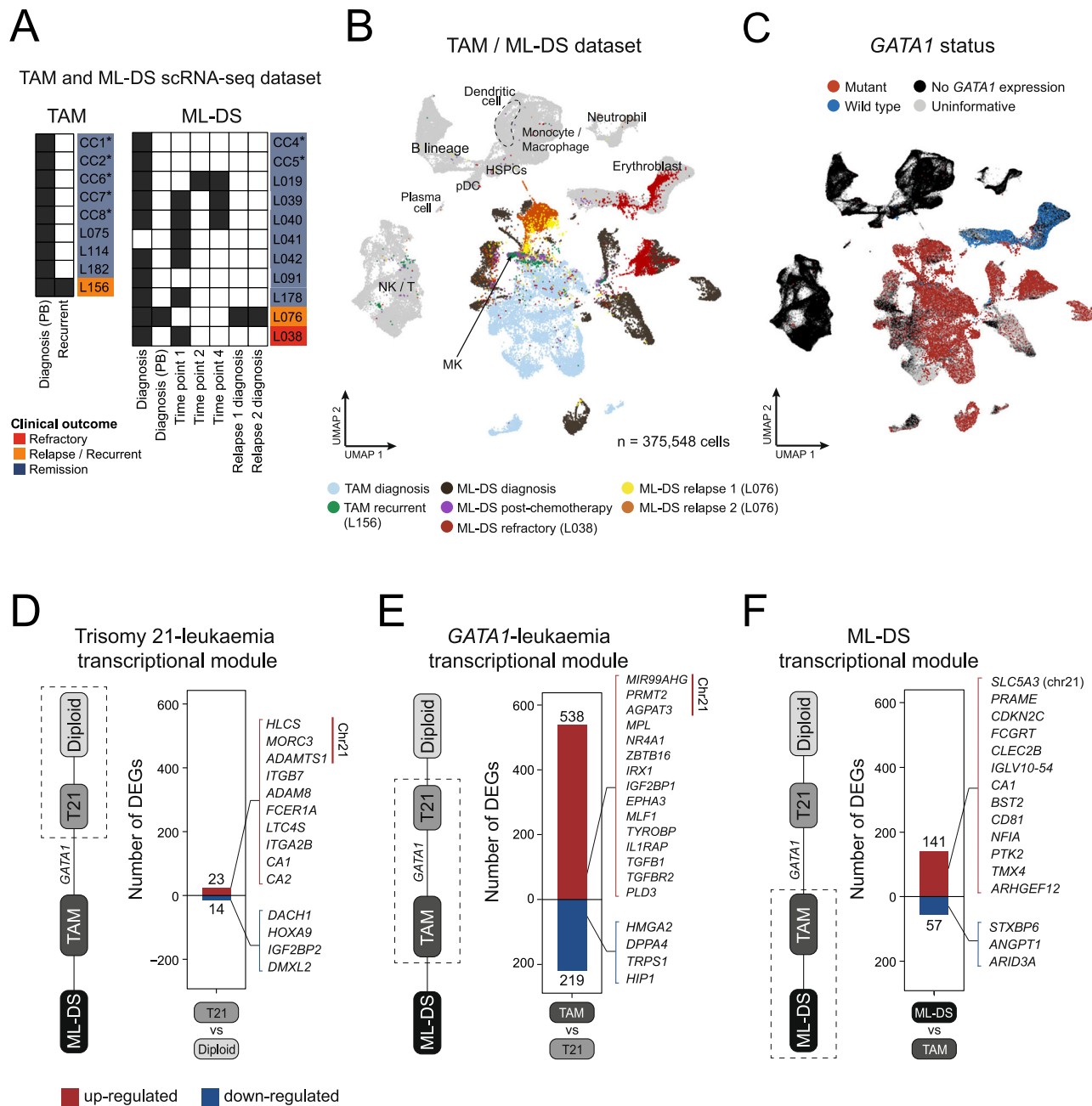

**Fig. 2 | Stepwise transcriptional progression towards ML-DS. A** TAM and ML-DS samples collected for scRNA-seq at various timepoints, from either peripheral blood (indicated by "(PB)") or bone marrow aspirates. Asterisks (*) indicate FACS-sorted samples to retain only leukaemic blasts for sequencing. **B** UMAP visualisation of the TAM / ML-DS scRNA-seq dataset, with cells (dots) coloured and/or labelled by cell-type categories. Non-leukaemic cells are in grey. HSPCs – haematopoietic stem and progenitor cells; MK – megakaryocyte; NK/T – natural killer cell / T cell; pDC – plasmacytoid dendritic cell. **C** UMAP visualisation of the TAM / ML-DS scRNA-seq dataset, highlighting cells with *GATA1* mutation (red dots) all within the leukaemic blast clusters. **D** Bar plot showing number of genes in the "trisomy 21 – leukaemia" transcriptional module (see Methods for definition). Red indicates up-regulated genes and blue indicates down-regulated genes in trisomy 21 foetal liver MEMP/MEP (and ML-DS blasts) compared to diploid foetal liver MEMP/MEP. Highlighted are selected up- and down-regulated genes from the module with known relevant functions reported in the literature. Chr21 denotes genes located on chromosome 21. The full gene module is detailed in Supplementary Data 3. **E** Bar plot showing number of genes in the "*GATA1* – leukaemia" transcriptional module, representing the contribution of *GATA1* mutations towards ML-DS transcriptome (see Methods for further details). Red indicates up-regulated genes and blue indicates down-regulated genes in TAM blasts (and ML-DS blasts) compared to trisomy 21 bone marrow MEP. Highlighted are selected up- and down-regulated genes from the module with known relevant functions reported in the literature. The full gene module is detailed in Supplementary Data 4. **F** Bar plot showing number of genes in the ML-DS transcriptional module, defined as differentially expressed genes between all diagnostic (treatment naive) ML-DS blasts against conventional (i.e. non-recurrent) TAM blasts (see Methods for further details). Red indicates up-regulated genes and blue indicates down-regulated genes in ML-DS blasts compared to TAM blasts. Highlighted are selected up- and down-regulated genes from the module with known relevant functions reported in the literature. The full gene module is detailed in Supplementary Data 5. Source data are provided as a Source Data file.

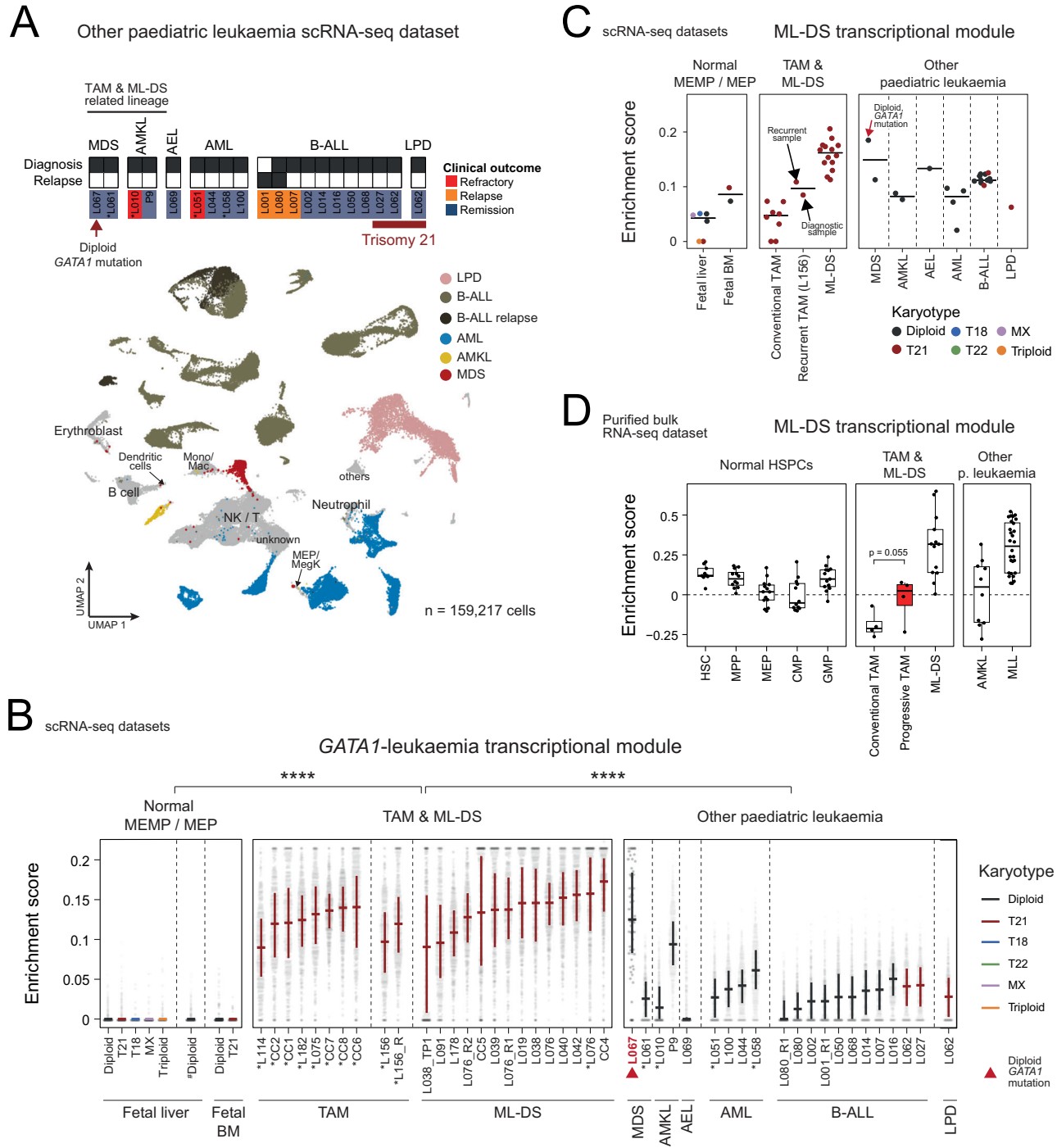

**A** Other paediatric leukaemia scRNA-seq dataset

**B** scRNA-seq datasets
*GATA1*-leukaemia transcriptional module

**C** scRNA-seq datasets — ML-DS transcriptional module

**D** Purified bulk RNA-seq dataset — ML-DS transcriptional module

Fig. 5B). These observations indicate that while *GATA1* mutations resulted in the most extensive transcriptional shift towards the ML-DS leukaemic state, a final transcriptional change is required to complete the evolution of ML-DS (Fig. 2D–F).

### Specificity of the trisomy 21 – TAM – ML-DS transcriptional progression

Having delineated the transcriptional changes associated with each sequential (stepwise) genetic lesion during the progression towards ML-DS, we set out to assess whether these signals are specific to ML-DS or shared amongst other leukaemias. To address this, we generated and examined a carefully curated dataset of 19 relevant leukaemias (159,217 single-cell transcriptomes) for comparison with TAM and ML-DS (as detailed in Table 1, Fig. 3A, Supplementary Fig. 6A), including:

rare cases of diploid myeloid neoplasm with and without *GATA1* mutation (Supplementary Fig. 6B, C), which enabled us to disentangle transcriptional effects of trisomy 21 and *GATA1*; morphologically and cell type related leukaemias (acute megakaryoblastic leukaemia–AMKL, acute erythroid leukaemia–AEL, other myeloid leukaemias); as well as lymphoid neoplasms with and without trisomy 21, to tease apart transcriptional effects of lineage (lymphoid versus myeloid) and trisomy 21. In addition, we included in our analyses relevant normal cell types from independent (published) datasets of foetal liver (diploid)[10] and foetal bone marrow (both diploid and trisomy 21)[11] cells. Finally, we generated an orthogonal bulk transcriptomic dataset for further validation, consisting of TAM ($n = 16$), ML-DS ($n = 13$), other myeloid leukaemias, including AMKL ($n = 10$) and *KMT2A*-rearranged leukaemia (also known as MLL-rearranged leukaemia; $n = 28$), along with flow-

**Fig. 3 | Specificity of the Trisomy 21 – TAM – ML-DS transcriptional progression.**
**A** Top: Other paediatric leukaemia samples collected for scRNA-seq. Asterisks (*) indicate peripheral blood (PB) samples; all others are bone marrow (BM) aspirates. Bottom: UMAP visualisation, with cells (dots) coloured and/or labelled by cell-type categories. Non-leukaemic cells are in grey. MEP – megakaryocyte-erythroid progenitor; Mono/Mac – Monocyte and Macrophages; NK/T – natural killer cell/T-cell.
**B** Enrichment scores of the "*GATA1*-leukaemia" transcriptional module (y-axis) across individual cells (grey dots) from normal foetal liver (# data from Popescu, D-M. et al., 2019[10]) and foetal BM (data from Jardine, L. et al., 2021[11]) MEMP/MEP; TAM/ML-DS blasts; and leukaemic blasts from other leukaemias. Leukaemic cells are grouped by patient, tissue (* for PB, otherwise BM aspirates), and timepoint (initial diagnosis unless indicated otherwise in the group name). For each group, cross bar vertical line shows IQR, and horizontal line indicate the median score, coloured by donor germline karyotype. Median scores in TAM/ML-DS
($n = 24$ samples from 19 individuals) are significantly higher than in normal MEMP/MEP ($n = 26$ samples across three foetal datasets), and other leukaemia

($n = 21$ samples) ($p < 0.0001$ for both comparisons, one-sided Wilcoxon rank sum test). No significant difference observed between TAM ($n = 10$ samples) and ML-DS ($n = 14$ samples) ($p > 0.05$, two-sided Wilcoxon rank sum test). R – recurrent; R1 – relapse 1 diagnosis; R2 – relapse 2 diagnosis; TP1 – timepoint 1 post-chemotherapy. **C** Per-sample median enrichment scores (dots) of the ML-DS transcriptional module (y-axis) across datasets as detailed in (**B**), coloured by donor germline karyotype. Black horizontal bars represent category-level medians. Black arrows highlight recurrent TAM samples with distinctly elevated scores compared to other conventional TAM cases. **D** Per-sample enrichment scores of the ML-DS transcriptional module (y-axis) in an independent bulk RNA-seq dataset. Boxes indicate first and third quartiles, central line represents the median, whiskers extend to 1.5×IQR. Near-significant difference observed between progressive ($n = 4$) and conventional ($n = 4$) TAM ($p = 0.055$, one-sided Wilcoxon rank sum test). HSPCs haematopoietic stem and progenitor cells; HSC haematopoietic stem cell; MPP, multipotent progenitor; CMP, common myeloid progenitor; GMP granulocyte-monocyte progenitor. Source data are provided as a Source Data file.

cytometry-sorted normal diploid haematopoietic progenitors (Table 1, Supplementary Data 1).

We calculated the enrichment score for each gene set of transcriptional changes we had defined to delineate the evolution of ML-DS from disomic liver cells. First, we examined imprints of the trisomy 21 transcription module (as defined in Fig. 2D). Amongst normal foetal MEP cells, these imprints were specific to trisomy 21. In neoplasms we found an enrichment in leukaemias of the MEP lineage (MDS with *GATA1* mutation, AMKL, AEL; Supplementary Fig. 7A). By contrast, strong enrichment of the *GATA1* transcription module (as defined in Fig. 2E) was confined to diploid MDS with *GATA1* mutation, indicating that the *GATA1* module does not depend on trisomy 21 (Fig. 3B). Of note, AMKL cancer cells exhibited a variable level of *GATA1* module (intermediate activity observed in P9 – a case of infant AMKL), which may reflect underlying biological differences between AMKL subtypes. Assessment in the independent bulk RNA-seq dataset validated these observations (Supplementary Fig. 7B, C).

Finally, assessing the ML-DS module (as defined in Fig. 2F), we found an enrichment across all leukaemias, including those from the B-lineage (Fig. 3C, Supplementary Fig. 8A). This suggests that the ML-DS module may represent a gene set widely dysregulated in leukaemia that is relevant beyond the transformation of TAM to ML-DS. Consistent with this notion, we identified established leukaemogenic genes, including *CD81*[40], *BST2*[18,41], *CD36*[42], *CLEC2B*[43], *PTK2*[44], within the ML-DS module (Supplementary Fig. 8B, Supplementary Data 5).

Of note, we also observed imprints of this cancer transcriptome in a child with "recurrent TAM" (L156, black arrows in Fig. 3C). This patient experienced re-expansion of the *GATA1*-mutant clone two months after initial cytarabine treatment (8% blasts by flow cytometry), and required a second course of cytarabine. The mutant clone then persisted at low levels (approximately 5%) before gradually clearing over the following 19 months. From a clinical treatment perspective, this case probably represents early ML-DS rather than TAM, and the high enrichment of the ML-DS transcriptional signature suggests that the module may capture neoplastic aggressiveness or early leukaemic transformation. Extending this observation into the independent bulk cohort, where 4 TAM cases eventually progressed to ML-DS (termed "progressive TAM"), we observed the same pattern (Fig. 3D). Although limited by small sample size, ML-DS enrichment scores were higher in progressive compared to conventional TAM samples, with a near-significant difference (one-sided t-test, $p = 0.055$), supporting the notion that the ML-DS transcriptomes may help delineate TAM cases at higher risk of progression. Furthermore, consistent with the idea that the ML-DS module captures broader leukaemic features, we observed enrichment in both *KMT2A*-rearranged leukaemias and a subset of AMKL cases in the bulk cohort (though not stratified by the genetic fusion event).

## Genetic and transcriptional evolution of progressive ML-DS

In a final analysis, we examined rare cases of progressive, ultimately fatal ML-DS: one child with two relapses (L076) and one child who developed refractory disease following the first course of chemotherapy (L038). With samples from all time points available, we were able to examine both the genetic and transcriptional evolution of progressive ML-DS. We inferred genetic evolution through whole-genome sequencing-based phylogenies using established analytical approaches[45]. We matched genetic clones with single-cell mRNA clusters by assessing allelic imbalances of cancer-defining copy number changes in mRNA reads, which is a precise and expression-independent approach for identifying cancer cells in single-cell mRNA sequencing data[46].

Following diagnosis with ML-DS, child L076 had two relapses after short-lived remissions (months) on each occasion. Phylogenetic analyses showed that the two relapses were derived from the diagnostic clone via a common precursor but then diverged in their genetic development, accruing mutations in parallel (Fig. 4A, Supplementary Data 6). The major clones of all three timepoints shared identical copy number profiles and key genetic drivers (Fig. 4B, Supplementary Fig. 9A, B), including homozygous loss of *CDKN2A*, a lesion associated with poor prognosis in ML-DS[20]. None of the additional substitutions or indels generated plausible driver events. Interestingly, the second relapse exhibited a mutational signature of prior treatment exposure (Fig. 4A). Consistent with the genetic distinction between initial disease and relapses, transcriptionally both relapses segregated clearly from the diagnostic clone (Fig. 4B). However, despite their subsequent parallel evolution—marked by the independent acquisition of private mutations—they remained transcriptionally similar to each other. This may indicate that these parallel clones retained a common transcriptional phenotype of progressive disease, or that both lineages converged on a common phenotype. Examining differential expression analyses between progressive disease and the initial diagnostic clone, revealed a range of expression changes that have been implicated in cancer cell survival or treatment resistance, such as upregulation of *FHL2*[47], *CXCL8*[48], *EPS8*[49] and *CD82*[50,51] (Fig. 4C, Supplementary Data 8). Overall, therefore, progressive disease in this child (L076) segregated genetically and transcriptionally from the initial diagnostic clone, which was underpinned by plausible expression changes without any specific mutation accounting for aggressive disease. By contrast, in child L038 who developed refractory disease after one course of chemotherapy, we identified a plausible progression event, namely deletion of *TP53* (through chromosome 17p deletion) which has previously been implicated as a driver of aggressive ML-DS[20] (Fig. 4D, Supplementary Fig. 9C). We found that progressive disease evolved from a clone with *TP53* deletion that had been present at diagnosis (Fig. 4E, Supplementary Fig. 9D, Supplementary Data 7). However, this

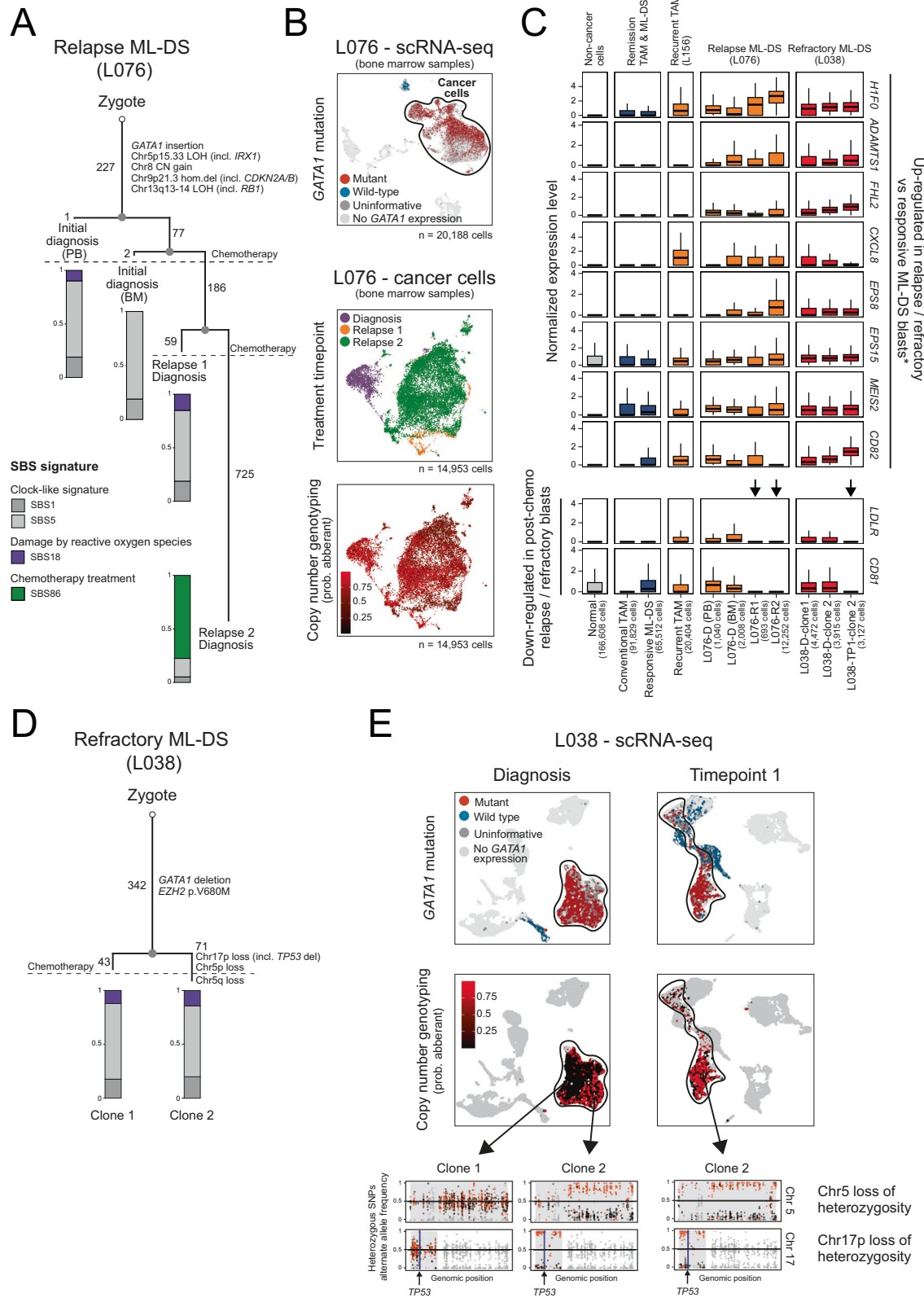

refractory clone had not been derived from the diagnostic clone but represented a parallel lineage—a subclonal population that co-existed with, and evolved independently of, the major clone present at diagnosis. The transcriptional difference between ML-DS cells with or without deletion of *TP53*, revealed a number of expression changes that plausibly contributed to disease progression, including some in common with expression changes of progression in the first child,

L076 (Fig. 4C, Supplementary Data 8). We observed specific down-regulation of genes such as *LDLR* and *CD81* in relapse/refractory blasts compared to the initial diagnostic clones. Importantly, genes underpinning these changes did not reside on the deleted chromosome 17p, although we were able to clearly observe transcriptional imprints of 17p deletion in refractory blasts (Supplementary Fig. 9E). Finally, we examined transcriptional commonalities between blasts from all

**Fig. 4 | Genetic and transcriptional evolution of progressive ML-DS.**
**A** Reconstructed phylogeny of clones in child L076 with two relapses, annotated with known driver mutations, copy number (CN) variants, and estimated number of single-nucleotide substitutions. Bar plots show the proportion of single-base substitution (SBS) signatures. Chr chromosome, PB peripheral blood, BM bone marrow. **B** UMAP visualisation of scRNA-seq data from bone marrow samples of L076. Top panel displays all cells (leukaemic and non-leukaemic), coloured by *GATA1*-mutation status. Middle and bottom panels show leukaemic cells, coloured by timepoint (middle) and probability of having CN aberrations (bottom). **C** Box plots showing single-cell normalised expression distribution for selected marker genes (rows) of relapse and/or refractory ML-DS blasts relative to treatment-naive ML-DS blasts from L076 and L038 (* – see Methods). Cells are grouped by identity and treatment outcome (x-axis): Normal – non-leukaemic cells from all TAM and ML-DS samples, Conventional TAM – leukaemic cells from all TAM samples except L156, responsive ML-DS – diagnostic blasts from all ML-DS samples except L076 and

L038, recurrent TAM – diagnostic blasts from L156. Boxes indicate first and third quartiles, central line represents the median, and whiskers extend to 1.5× IQR. D – Diagnosis; R1 – relapse 1 diagnosis; R2 – relapse 2 diagnosis; TP1 – timepoint 1.
**D** Reconstructed phylogeny of clones in child L038 with treatment-refractory disease, annotated with known driver mutations, CN variants, and the estimated number of single-nucleotide substitutions. SBS signature profiles are shown in bar plots using the same colour scheme as in (**A**). **E** UMAP visualisation of L038 scRNA-seq data. All cells (dots) from both timepoints are shown. Cells are coloured by *GATA1* mutation status (top) and the probability of having CN aberrations (middle). Bottom panel shows the aggregated B-allele frequency of heterozygous single-nucleotide polymorphisms (SNPs) across clones from each timepoint. Each dot represents a heterozygous SNPs along chromosomes 5 and 17 (x-axis). Orange dots indicate alternate alleles on major chromosome; black dots indicate alternate alleles on minor chromosome. CN segments are grey-shaded regions. Source data are provided as a Source Data file.

samples of these two children which showed that, irrespective of disease stage, blast transcriptomes retained *GATA1* induced gene expression (Fig. 2E), underscoring our proposition that they may represent a common vulnerability across the ML-DS disease spectrum.

## Discussion

We studied the genetic and transcriptional evolution of ML-DS directly from natural variation, which enabled us to delineate transcriptional effects of individual genetic steps and measure their contribution to the ML-DS transcriptome. A key finding of our analyses was that *GATA1* induced transcriptional changes predominated the entire disease spectrum, including relapse/refractory disease. The genetic changes that underpin the progression from TAM to ML-DS do not override or erase the *GATA1* transcriptome. This indicates that *GATA1*-mediated expression remains relevant across all disease stages and may thus represent a persistent therapeutic vulnerability.

The strength of our work lies in leveraging natural variation from primary disease samples as the basis of our investigation. High-throughput single-cell mRNA sequencing enabled us to extract relevant cell populations for precise expression analyses, distiling the transcriptional effects of genetic variation. We studied several individuals with TAM and ML-DS, including cases of progressive ML-DS, enabling us to derive patient-overarching effects that we further validated in an independent bulk transcriptomic dataset. A key limitation of our approach is its dependence on natural variation. Accordingly, some of the comparators in our analyses are individual cases of conditions that are vanishingly rare, such as a foetus with whole genome triploidy or a disomic child with *GATA1*-driven MDS. While single cases certainly cannot capture the entire spectrum of transcriptional variation within each condition, they nonetheless offer critical reference points for evaluating the specificity of the transcriptional signatures associated with the genetic aberrations along the development of ML-DS. Furthermore, even in these cases, single-cell transcriptomics delivered multiple readouts, which substantiates our findings at the level of individual patients. Nevertheless, larger cohorts of patients with these rare conditions will be necessary to validate these observations and further refine our understanding of mutation-specific transcriptional consequences.

Previous work has explored the functional significance of each genetic aberration along the development of ML-DS through in vitro assays using primary human haematopoietic stem and progenitor cells, mouse models or xenotransplantation in mice[1,32,38,39,52–55]. Here, we examined the contribution of each genetic step in the transcriptional evolution of ML-DS and showed that *GATA1* mutations account for most of the ML-DS transcriptome. This held true in progressive ML-DS in two children. Despite complex phylogenetic relationships among cancer clones and transcriptional changes at each step, the *GATA1* transcriptome continued to predominate in ML-DS-blasts. *GATA1*-

induced changes were independent of constitutional trisomy 21 and specific to *GATA1*-mutated neoplasms, in keeping with previous reports suggesting that trisomy 21 is not essential for leukaemogenesis[32,56]. This observation would explain why ML-DS can only be generated via *GATA1* mutations, as well as clarifying rare cases of TAM/ML-DS-like neoplasms with *GATA1* mutation in children without Down syndrome[56]. Unlike the highly disease-specific effects of *GATA1* mutation, ML-DS-defining transcriptional changes represented a generic leukaemogenic transcriptome, underscoring the fact that ML-DS is a cancer. Given the central contribution of *GATA1* mutation-associated transcriptional changes to ML-DS, it is conceivable that therapeutic interference with these changes may disintegrate and, thus, treat the ML-DS transcriptome.

Owing to meticulous tissue banking efforts, we were able to study rare clinical phenotypes of progressive TAM and ML-DS. Recurrent TAM is a largely cryptic disease which clinically is indistinguishable from ML-DS and often, on clinical grounds, will eventually be treated as ML-DS. No molecular differences between progressive and non-progressive TAM have been described to date, and there are no effective risk stratification strategies[57–60]. Remarkably, both single-cell and bulk transcriptomes of progressive TAM were transcriptionally distinct from conventional (i.e., non-recurrent) TAM and more closely resembled ML-DS. This indicates that enriching the ML-DS transcriptional signature, including markers such as *CD81*, *BST2*, *CD36*, and *CLEC2B*, may identify potential biomarkers enabling the prospective identification of progressive TAM. Validation in larger, longitudinal cohorts will be essential to account for inter-patient heterogeneity and to refine the most robust, clinically actionable components of this transcriptional signature. While the transcriptional module as a whole may serve as a composite biomarker, further investigation is warranted to identify individual genes or gene combinations within the module that possess translational relevance.

Our quantitative analysis in primary human samples, integrated with DNA analyses, details a unified landscape of the transcriptional evolution underpinning ML-DS. Some of our findings, such as transcriptional features of progressive TAM or the pre-existence of a refractory clone at diagnosis in one child, may be clinically useful. Pursuing these clinical hypotheses further will require large-scale, international prospective investigations. At the same time, our work outlines a broadly applicable analytical approach for distiling transcriptional effects of individual mutations directly from human cancer cells and natural variation rather than model systems.

## Methods

### Sample acquisition, ethics and patient consent
Foetal liver samples were obtained from the MRC–Wellcome Trust – funded Human Developmental Biology Resource (HDBR; http://www.hdbr.org[61]) with appropriate maternal written consent and approval

from the North East – Newcastle & North Tyneside 1 Research Ethics Committee. HDBR is regulated by the UK Human Tissue Authority (HTA; www.hta.gov.uk) and operates in accordance with the relevant HTA Codes of Practice.

Leukaemia samples from bone marrow aspirates or peripheral blood were collected through studies approved by the UK NHS research ethics committee. The tissue sources are: Great Ormond Street Hospital for Children diagnostic archives, as approved by a National Research Ethics Service Committee (London Brent, reference 16/LO/0960); and the AML Berlin-Frankfurt-Münster (BFM) study group. Bulk transcriptomic processed data with approval from the Goethe University Frankfurt (Ethics # 2021-341 and # 2023-1565). Ethical approval was obtained from the London – Brent Research Ethics Committee, the University Duisburg-Essen (Ethics #17-7462), Hannover Medical School (Ethics #6741 M) and Martin-Luther-University Halle-Wittenberg (Ethics #2019-103). Relevant clinical information of the patient cohort is included in Supplementary Data 1. Informed consent to participate in research was obtained from all patients or their legal guardians as stipulated by the study protocols.

### TAM / ML-DS blast enrichment by flow cytometry
TAM and ML-DS samples obtained from the AML Berlin-Frankfurt-Münster (BFM) study group were FACS-sorted to obtain pure leukaemic blasts prior to the 10X workflow (Supplementary Fig. 10, Supplementary Data 1). Mononuclear cells were enriched with density gradient centrifugation and subsequently FACS sorted. Blast was defined as CD45dim/CD33 + /CD117 + /CD7 + /CD11a-.

### 10X single-cell RNA sequencing (scRNA-seq)
Fresh foetal livers were dissociated into single-cell suspensions. For the leukaemia samples (bone marrow aspirates or peripheral blood), peripheral blood mononuclear cells were prepared by density centrifugation using Lymphoprep (Stemcell) according to manufacturer's instructions, also generating single-cell suspensions. The suspension was passed through a 70μm cell strainer (Falcon) and washed with PBS. If necessary, live cell enrichment using a Dead Cell Removal kit (Miltenyi Biotec) and red blood cell removal using the eBioscience 10X RBC Lysis Buffer (Multi-species) were performed as per the manufacturer's instructions. For some foetal liver samples, cells are further sorted into CD45+ and CD45- fractions using magnetic beads (detailed in Supplementary Data 1). All enriched live cells were washed and counted using a hemocytometer with trypan blue; single cell suspensions were adjusted to 1000 cells/ul accordingly. Cells were loaded onto the Chromium 10X controller as per the standard protocol of the Chromium Single Cell 3' Reagent kits (V3 chemistry) or 5' Reagent kits (v2 chemistry) in order to capture between 7000 cells/chip position (detailed in Supplementary Data 1). All the following steps were performed according to the standard manufacturer protocol. Post GEM-RT clean-up, cDNA amplification, and 5' gene expression library construction were carried out according to the manufacturer's instructions. The resulting libraries were sequenced on the Illumina Novaseq 6000 platform, aiming for an average of 300,000 reads per cell.

### Quality control and preprocessing of scRNA-seq data
Raw sequencing data were processed and mapped to the reference genome (GRCh38 1.2.0 or GRCh38 2020-A as detailed in Supplementary Data 1), using Cell Ranger pipeline[62]. The filtered count matrix, outputted by Cell Ranger, was then further QC'ed using Seurat (v4.0.1)[63] in R (v4.0.4). Cells with <300 genes, <500 UMIs, or mitochondrial fraction exceeding 30% were removed. Scrublet (v0.2.3)[64] was used to identify doublets. Cells were excluded if identified as doublets by Scrublet, or having a doublet score > 0.5. Ambient mRNA contamination was removed with SoupX (v1.6.1)[65]. High resolution clusters (resolution=10) with >50% cells failing QC were also excluded.

The data were log-normalised and scaled, and principal components were calculated using highly variable genes, following the standard Seurat workflow. Louvain clustering was performed (resolution = 1), and a uniform manifold approximation and projection (UMAP) was calculated, using the top 75 principal components. No integration or batch-correction methods were performed to preserve biological variation across different karyotypes and cancer samples.

### Bulk mRNA sequencing and processing
RNA was isolated from cells using the Quick-RNA Miniprep Kit (Zymo Research). Paired-end libraries with 2 × 50, 75, 100 and 150 bp reads were prepared from the extracted RNA using the TruSeq Stranded total RNA LT Sample Prep (RiboZero Gold, Illumina) using Illumina Methodology by Novogene Company, Ltd. Raw sequencing reads were trimmed with fastp (v0.24.0)[66] and aligned to GRCh38 using STAR (v2.6.0)[67]. Mapped reads were annotated using Ensembl v.108. Gene expression levels in transcripts per million (TPM) were quantified from the mapped reads using salmon (v1.10.3)[68].

### Whole genome sequencing
Whole genome sequencing (WGS) was performed on the foetal tissues with abnormal karyotypes, and those from two cases of aggressive ML-DS: one relapsed twice, and one refracted to standard chemotherapy (details in Supplementary Data 1). Bulk DNA sequencing was performed as previously described[69]. In brief, DNA was extracted using the AllPrep DNA/RNA/Protein Mini Kit (QIAGEN) following the standard protocol, and short insert (~500 bp) genomic libraries were generated. Finally, 150 bp paired-end sequencing was conducted on the Illumina NovaSeq 6000 platform according to Illumina's standard library generation protocols (with PCR). The average sequence coverage was at least 30X per sample (details in Supplementary Data 1).

Raw DNA sequencing data were aligned to the GRCh38 (Ensembl 103) reference genome using the Burrows-Wheeler algorithm (BWA-MEM)[70].

### Sample-level genotyping
To ensure that sequencing data (both WGS and scRNA-seq) from the same individuals are correctly labelled, sample-level genotyping was performed using the *matchBAMs* function from the R package *alleleIntegrator*[46].

The karyotype of each donor was also confirmed. For those with WGS data (see Supplementary Data 1), we compared the coverage of affected chromosome(s) to the rest of the genome as follows:
- Single-chromosome trisomy: coverage of the trisomic chromosome is approximately 1.5 times higher than that of other disomic chromosomes (other copy number regions were excluded, if any).
- Monosomy X: There is no coverage on chromosome Y (confirming that the individual is a female), and the coverage of chromosome X is half that of other disomic chromosomes.
- Whole genome trisomy: We assessed the distribution of the alternate-allele frequency at heterozygous single-nucleotide polymorphisms (SNPs), which showed a bimodal distribution with peaks around 1/3 and 2/3.

For cases with only scRNA-seq data, we first combined all scRNA-seq samples from the same individual and identified the individual-specific heterozygous SNPs using the R package *alleleIntegrator*[46]. We then calculated the alternate/B-allele frequency (BAF) at heterozygous SNPs, aggregated across all high-quality sequencing reads (mapping quality >= 200, base quality >= 20, minVarQual = 225) on each chromosome. By assessing BAF distributions for each individual, we confirmed that trisomic chromosomes exhibit a bimodal BAF distribution with peaks around 1/3 and 2/3, while disomic chromosomes show a unimodal distribution around 0.5.

## Cell type annotation

**Foetal liver scRNA-seq dataset.** A semi-automated cell type annotation using a label transfer approach was employed. Using Celltypist[71], a logistic regression model was trained on the reference scRNA-seq atlas of the human foetal livers (Popescu, DM., Botting, R.A., Stephenson, E. et al., 2019[10]). The model was then used to calculate predicted similarity scores for individual cells in our query foetal liver dataset against each reference class (i.e., different cell types in the reference dataset) (Supplementary Fig. 1D). Cells were assigned the label of the class with the highest positive similarity score. Annotation was further refined by manually assessing the expression of well-established cell-type-specific marker genes (Supplementary Fig. 1C) in high-resolution Louvain clusters.

To quantify the proportion of different haematopoietic lineages in the foetal livers of different karyotypes, we first excluded all non-haematopoietic cells, including: endothelial cells, fibroblasts, hepatocytes, cholangiocytes, mesothelial cells, neurons, and trophoblasts. The remaining cells were grouped by donor ID and corresponding lineages as follows:

- HSC/MPP
- Megakaryocyte – Erythroid – Mast cell lineage: MEMP/MEP, early megakaryocytes, megakaryocytes, early/mid/late erythroblasts, mast cells.
- B lineage: LMPP/ELP, pro B cells, pre B cells, B cells.
- Myeloid lineage: CMP/GMP, DC1, DC2, pDC, pro-monocytes, monocytes, macrophages, Kupffer cells, pro-myelocytes, myelocytes.
- NK/T lineage: ILC precursors, natural killer/T cells.

The proportion of each lineage in each donor was calculated as the number of cells in the lineage divided by the total number of haematopoietic cells from that donor.

Comparisons of the lineage proportions between those with trisomy 21 samples and diploid foetal livers were performed using donor-level proportions with a one-tailed Wilcoxon rank-sum test (using the *wilcox.test* function in R), treating each donor as a biological replicate.

**Leukaemia scRNA-seq dataset.** A similar approach was used to annotate both leukaemia scRNA-seq datasets (TAM/ML-DS and other leukaemias), where automated label transfer was followed by manual assessment of canonical cell type-specific marker genes. However, as these datasets contain neoplastic cells that may not closely resemble normal cellular transcriptomes, we employed a logistic regression method previously detailed in Young et al., 2018[69]. Specifically, a logistic regression model with elastic net regularisation (alpha = 0.99) was trained on our diploid foetal liver scRNA-seq data (reference dataset), using the *cv.glmnet* function from the *glmnet* R package. This model was applied to calculate the predicted similarity scores for individual cells in the leukaemia datasets against each reference class (i.e., different cell types in the reference dataset) (Supplementary Fig. 3C). To allow for the possibility that some query cells may not match strongly to any of the reference cell types, softmax normalisation was not used. Cells were assigned the label of the class with the highest positive similarity score. Annotation was further refined by manually assessing the expression of well-established cell type specific marker genes (Supplementary Fig. 3D) in high-resolution Louvain clusters.

## Identification of *GATA1* mutated neoplastic cells in TAM / ML-DS scRNA-seq data

**Genotyping for the presence of GATA1 mutations in single-cell transcriptomes.** The exact *GATA1* mutation(s) acquired in each TAM / ML-DS case was obtained from clinical history and confirmed with our WGS analyses where possible. Next, we examined the scRNA-seq reads from each sample, using samtools[72] to identify high quality reads overlapping the *GATA1* mutation position +/− 70 bases. These reads were sorted into *GATA1* mutant or wild-type reads based on the nucleotide sequence; and assigned to their corresponding cell barcodes. The number of mutant or wild-type *GATA1* reads per cell was tallied. It is expected that only one allele of *GATA1* is expressed in a given cell, as *GATA1* is located on chromosome X outside the PAR region, combined with X-inactivation in females. Therefore, cells were classified as mutant (and therefore neoplastic cells) if they expressed only the *GATA1* mutant allele, and normal if they exclusively expressed the wild-type allele. For cells with both wild-type and mutant reads (likely due to doublets or ambient RNA contamination), cells were classified as mutant if they had at least five more mutant reads than wild-type reads, or wild-type if the reverse was true. Otherwise, they were considered ambiguous.

## Differential gene expression analysis

All differential gene expression analyses outlined below employed a donor-level pseudo-bulk approach. Donor-level pseudo-bulks were generated by aggregating raw counts per gene across all relevant cells from each donor. Pseudo-bulks with fewer than 30 cells were excluded, along with genes on chromosome Y. DESeq2[73] was used to perform differential gene expression analysis, following the workflow outlined in the vignette [https://bioconductor.org/packages/devel/bioc/vignettes/DESeq2/inst/doc/DESeq2.html]. In brief, a negative binomial generalised linear model was fitted to estimate the effect of independent variables on gene expression levels. Wald statistics with Benjamani-Hochberg correction for multiple hypothesis testing were used to identify genes with significant differential expression (i.e., genes with log2 fold-change (log2FC) of expression level significantly different from zero). The log2FC for each gene was extracted and subjected to shrinkage correction to improve accuracy for genes with high dispersion. Genes were identified as significantly differentially expressed (DEGs) using the following criteria unless stated otherwise:

1. Adjusted p-value < 0.05
2. Absolute log2FC >= 0.5
3. Expressed in >= 10% of cells across all samples from either group

## Transcriptional consequences of different abnormal karyotypes on foetal liver cell types

To assess the transcriptional impact of different abnormal karyotypes on foetal liver cell types, we performed pseudo-bulk differential expression analyses comparing each cell type from individual abnormal karyotypes with the diploid counterpart (Fig. 1E, F; Supplementary Fig. 2; Supplementary Data 2). Due to low cell numbers, some cell types were grouped together:

- Megakaryocyte: early megakaryocytes and megakaryocytes.
- Myelocyte: pro-myelocytes and myelocytes.
- Monocyte: pro-monocytes and monocytes.
- B cell progenitors ("B.cell.prog"): LMPP/ELP; pro B cells; and pre B cells.
- NK/T: ILC precursors and natural killer / T cells.

To estimate the sample-specific size factor, the *estimateSizeFactors* function was used with *controlGenes* parameter specified to only genes on disomic chromosomes. For the triploid vs diploid comparison, *controlGenes* is specified as genes on chromosome X outside the PAR region (which is known to escape X-inactivation), with the assumption that despite being triplicated, the expression level of these genes is least likely to be affected compared to diploidy due to the X-inactivation mechanism in females. The model design used was "Gene_expression ~ Karyotype + Assay".

Genes were considered to be expressed by a given cell type if they were expressed in >= 10% diploid cells of that cell type.

For each karyotype, the genome-wide "off-target" transcriptional impact in each cell type was defined as the proportion of DEGs residing outside the affected chromosome(s) (Fig. 1E).

## Derivation of the trisomy 21 – leukaemia transcriptional module

We defined the "trisomy 21 – leukaemia" transcriptional module as the transcriptional changes observed in trisomy 21 foetal liver MEMP/MEP cells compared to their diploid counterpart that is contained within the ML-DS transcriptome.

First, the ML-DS transcriptome is defined as DEGs between diagnostic ML-DS blasts (from bone marrow samples only) compared to diploid foetal liver MEMP/MEP cells. To minimise potential confounding artifacts, only diploid foetal liver samples sequenced using the same 10X reagent kit (Chromium Single Cell 5′ Reagent kits – v2 chemistry) as the ML-DS samples were included. The model design used was "Gene_expression ~ Group". We then examined which of these DEGs were already significantly perturbed in the same direction in trisomy 21 foetal liver MEMP/MEP. We performed a second pseudobulk analysis comparing expression levels of these genes in cells with trisomy 21 and diploid cells. Here, all relevant foetal liver samples sequenced with either 5′ or 3′ reagent kit were included. To account for the effects of different reagent kits, the model design used was "Gene_expression ~ Karyotype + Assay", where "Assay" is a two-level factor representing reagent kits. Genes which met criteria (1) above were considered to be significantly differentially expressed in trisomy 21 foetal liver MEMP/MEP, and thus included in the "trisomy 21 – leukaemia" transcriptional module. Criteria (2) and (3) were excluded here as we wanted to retain all relevant changes in expression level, including very subtle perturbations represented by low log2FC values. Full results from both analyses can be found in Supplementary Data 3.

## Derivation of the GATA1 – leukaemia transcriptional module

The "GATA1 – leukaemia" transcriptional module reflects the transcriptional consequences of GATA1 mutations observed in TAM blasts compared to trisomy 21 MEP cells, which are contained within the ML-DS transcriptome.

We first performed a pseudobulk differential expression analysis comparing diagnostic bone marrow ML-DS blasts with normal trisomy 21 MEP cells. More specifically, the latter cells were obtained from post-treatment remission bone marrow samples of four different ML-DS patients (L019, L038, L040, L041) with the highest number of captured MEP cells. DEGs were identified using all three criteria outlined above. We then tested for the differential expression of these genes between conventional (i.e. non-recurrent) TAM blasts and the same set of trisomy 21 MEP cells. Genes which met criteria (1) were considered to be significantly differentially expressed in TAM blasts as transcriptional changes induced by GATA1 mutations, and thus included in the "GATA1 – leukaemia" transcriptional module. Full results from both differential expression analyses can be found in Supplementary Data 4.

To better understand the "GATA1 -leukaemia" transcriptional module, we examined the expression of each gene across normal haematopoietic cells from our diploid foetal liver scRNA-seq dataset. The average expression per gene was calculated for each cell-type category using the AverageExpression function from the Seurat package[63]. Genes with no detectable expression in any haematopoietic cell categories were excluded from the heatmap shown in Supplementary Fig. 5A.

## Derivation of the ML-DS leukaemia transcriptional module

The "ML-DS leukaemia" transcriptional module captures the transcriptional differences between the leukaemic state of ML-DS and the pre-leukaemic state of TAM blasts. A pseudobulk differential expression analysis was performed comparing ML-DS cancer cells from diagnostic bone marrow samples against conventional TAM blasts. Genes were considered to be DEGs if all three criteria outlined above were met. Full results can be found in Supplementary Data 5.

Grouping all ML-DS cases into one category may have obscured between-patient heterogeneity, which is known to be a prominent feature across cancer types. To address this, we performed a second analysis comparing the conventional TAM group against individual ML-DS cases. DEGs for each comparison were extracted using the above 3 criteria. This analysis revealed a large number of transcriptional perturbations in ML-DS compared to TAM which are individual-specific (Supplementary Fig. 5B).

## Characterisation of the transcriptional signature of progressive ML-DS

To investigate the transcriptional changes potentially associated with progressive ML-DS and chemotherapy resistance, we specifically looked for markers associated with chemotherapy resistance in the progressive blasts in L076 and L038 independently, using the Find-Markers function from the Seurat package[63]. For L076, we compared progressive cancer cells from relapse 1 and relapse 2 against those from the initial diagnostic samples. For L038, we compared cancer cells with TP53 loss in timepoint 1 samples against cancer cells without TP53 loss in the initial diagnostic samples. Full results can be found in Supplementary Data 8.

## Transcriptional module enrichment analysis

The trisomy 21 – leukaemic gene module was composed of all genes derived from the above analysis. The GATA1 – leukaemic gene module and ML-DS leukaemic gene module from the above analyses were further refined to retain only genes expressed in >= 20% more cells in the group with higher expression. The enrichment score for each gene module (composed of both up- and down-regulated genes) in single-cell RNAseq data was calculated using the function AddModuleScore_UCell from the R package UCell[74] (v1.3.1). For the bulk transcriptome dataset, the enrichment score was computed using the function simpleScore from the R package singscore[75] (v1.10.0).

## Phylogenetic reconstruction of progressive ML-DS

**Somatic variants calling.** We investigated the genetic evolution of relapse and refractory ML-DS in child L076 and child L038 respectively. For both cases, the genetic phylogenetic relation was reconstructed based on somatic single-nucleotide variants (SNVs, also referred to as substitutions). SNVs in both the diagnostic and TP1 WGS samples were called using the CaVEMan algorithm[76] (v1.18.2), and short insertions/deletions were called using Pindel algorithm[77] (v3.10.0) in an unmatched analysis of each sample against an in silico normal human reference genome. In addition to the inbuilt QC filters, we further applied a series of stringent filters to remove low quality variants and germline variants, following the detailed workflow outlined in Coorens et al., 2019[45]. Briefly, we only retained variants with a high median alignment score of supporting reads (ASMD > = 140), and required that fewer than half of the reads were clipped (CLPM = 0), as well as not falling within ten base pair of a deletion or insertion called by Pindel. We then recounted across both samples the variant allele frequency of all substitutions with a cut-off for base quality of 25 and read mapping quality of 30. Variants were also filtered out if they were called in a region of consistently low depth across both samples (excluding copy number segments). Next, to filter out germline substitutions, we fitted a binomial distribution to the combined read counts across all samples per SNV site, and applied a one-sided exact binomial test to calculate the probability that the SNV is consistent with being a germline variant. For the remaining variants, we calculated the site-specific error rates by interrogating the same sites in a panel of normal blood samples, using a beta-binomial model derived from the Shearwater variant call[78]. Variants which are indistinguishable from the background error rates were removed. All variants which passed all filters were visually inspected using the genome browser Jbrowse[79] to exclude further sequencing or mapping artefacts. The final list of somatic substitutions can be found in Supplementary Data 6 for L076 and Supplementary Data 7 for L038.

## Single-base substitution signature analysis

The single-base substitution profile of each WGS sample from L076 and L038 was further deconvolved into linear combinations of known COSMIC reference signatures (v3.4) using the R package mSigAct[80] (v3.0.1). The set of reference signatures provided was: SBS1, SBS5, SBS13, SBS15, SBS18, SBS31, SBS35, SBS44, SBS84, and SBS86. Signature exposures per sample can be found in Fig. 4A, C.

## WGS-derived copy number aberrations

We identified copy number aberrations in each WGS sample from L076 and L038 using the COBALT, AMBER, and PURPLE toolkit developed by the Hartwig Medical Foundation, following their documentation available on github: https://github.com/hartwigmedical/hmftools. Briefly, the BAF of heterozygous single-nucleotide polymorphism (SNP) sites was computed using AMBER (v3.9), and read depth ratios were calculated with COBALT (v1.14.1), both in "tumour-only" mode. BAF information and read-depth ratios were integrated to estimate the purity, ploidy and copy number profile of each tumour using PURPLE[81] (v3.8.4).

## Detection of copy number aberrations in scRNA-seq data

Using the R package alleleIntegrator[45] (v0.9.1), we examined the copy number status of individual leukaemic blasts from patient L076 (relapse ML-DS), L038 (refractory ML-DS) and L067 (diploid MDS with *GATA1* mutation). Heterozygous SNPs were identified from WGS data from each individual, and phasing of heterozygous SNPs across copy number segments with allelic imbalance was performed. The number of scRNA-seq reads supporting the major/minor allele across each copy number segment in each cell was calculated. Finally, the posterior probability of each cell harbouring the altered or normal copy number state for each copy number segment was calculated.

## Reporting summary

Further information on research design is available in the Nature Portfolio Reporting Summary linked to this article.

## Data availability

WGS and single-cell mRNA sequencing data has been deposited in the European Genome-Phenome Archive (EGA) under the accession codes EGAD00001015453 (WGS; https://ega-archive.org/datasets/EGAD00001015453) and EGAD00001015452 (scRNA-seq; https://ega-archive.org/datasets/EGAD00001015452). Processed counts for bulk transcriptomic data are available as Supplementary Data 9; the corresponding raw sequencing data are available through the Zenodo repository at https://doi.org/10.5281/zenodo.19046231. Published foetal liver (Popescu, D-M. et al., 2019[10]) and foetal bone marrow (Jardine, L. et al., 2021[11]) datasets data were downloaded from https://developmental.cellatlas.io. Source data are provided with this paper.

## Code availability

All code used to reproduce the analysis and figures described in this manuscript is available at https://github.com/miktrinh/ML-DS.

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

## Acknowledgements

This study was funded by the Wellcome Trust (institutional grant to the Wellcome Sanger Institute (WT206194), and personal fellowship to S.B. (223135/Z/21/Z)). This study was supported by "Hilfe für krebskranke Kinder Frankfurt e.V." as part of the C$^3$OMBAT-AML consortium, the German Research Foundation (DFG, KL 2374/8-1), the Blood Cancer United (SCOR #7039-25), and the European Research Council (ERC) under the European Union's Horizon 2020 Research and Innovation Programme (grant agreement No. 714226). This research was also supported by the NIHR Cambridge Biomedical Research Centre (NIHR203312). The views expressed are those of the authors and not necessarily those of the NIHR or the Department of Health and Social Care. Additional funding was received from the Wenner-Gren Foundations (personal fellowship to A.W.), and The Little Princess Trust (personal fellowship to T.D.T.). We are indebted to the children and families who participated in this study.

## Author contributions

S.B., J.B., and J.H.K. conceived and jointly supervised the study. M.K.T. performed overall bioinformatic data analysis and interpreted the results, aided by M.D.Y., L.J., K.Sch., A.W., N.D.A., H.J.W., T.D.T, A.H., J.G.D., S.A. and D.O.C. H.I., R.T., C.P., A.O., T.O., D.Z., L.M., E.P., E.R.R., A.H., K.Str., and S.A. collected and processed the samples for sequencing experiments. S.B. and M.K.T. wrote the manuscript. All authors have read and approved the manuscript.

## Competing interests

J.H.K. has advisory roles for Pfizer, Boehringer, Roche and Jazz Pharmaceuticals.
