## [Transparent Peer Review file · Nature Communications]

Single cell transcriptional evolution of myeloid leukaemia of Down syndrome

Corresponding Author: Professor Sam Behjati

Version 0:

Reviewer comments:

Reviewer #1

(Remarks to the Author)

Development of myeloid leukemia in children with DS is a stepwise process that evolves from fetal liver cells with trisomy 21 to transient abnormal myelopoiesis which is defined by the acquisition of a GATA1 mutation, and finally to myeloid leukemia upon mutations in other genes such as members of the cohesion complex. This paper provides the first single cell RNA-seq analysis of this leukemia. The authors leverage many samples from the various stages of disease to define the features of this step wise leukemia development. While some of the data, such as an enrichment of megakaryocyte/erythroid lineage with decreased B-cell lineage, are confirmatory, other findings are novel and significant. Also, the comprehensive gene expression details may facilitate novel treatment strategies to prevent progression and better treat relapsed disease. The paper also provides a rich and significant dataset.

A minor limitation is that data for some of the disease subtypes are derived from a single patient. This includes comparisons of T21 with the T22 and genome wide triploid specimens as well as only one MDS sample with a GATA1 mutation (and one without), a single acute erythroid leukemia sample, and a single lymphoproliferative disease specimen. While additional samples would be helpful, these subtypes provide interesting comparisons. I'd like to see a bit more discussion of the points on lines 313-314 which mentions the possibility of using some of the data as potential biomarkers. Are there any strong candidates that can be noted here? Also, on line 203 it would be helpful for the authors to specify that the "other myeloid leukemias" are composed of AMKL and MLL-rearranged cases (this is listed in Table 1 but would be helpful to have it mentioned in the main text). Finally, what are the karyotypes of the MDS cases with and without trisomy 21 but with and without the GATA1 mutation, and do the AMKL cases include different translocations? Is there heterogeneity among the AMKLs if they have different fusion proteins?

Reviewer #2

(Remarks to the Author)

Trinh and his colleagues performed single-cell mRNA sequencing on fetal livers with various constitutional karyotypes, including trisomy 21, as well as on primary patient samples with myeloid neoplasms related to Down syndrome and other types of leukemia. The authors demonstrated the stepwise transcriptional progression to myeloid leukemia of Down syndrome (ML-DS), including the effects of trisomy 21 on fetal hepatic hematopoiesis, the development of transient abnormal myelopoiesis (TAM) caused by *GATA1* mutations, the progression of TAM to ML-DS through additional mutations, and the development of progressive, fatal ML-DS. They showed that GATA1-induced transcriptional changes predominate throughout the disease spectrum, including relapsed/refractory disease.

This technically sound work adds to our current state of knowledge regarding the pathophysiology of ML-DS. The following minor suggestions could further improve the manuscript.

1. It is very interesting that the ML-DS transcriptional module enrichment scores for recurrent TAM and progressive TAM were intermediate between those for conventional TAM and ML-DS. The following details should be clarified:
 - a. Please provide a more detailed description of the clinical course of recurrent TAM patient (L156).
 - b. Are there statistically significant differences in the median enrichment score per-sample between progressive TAM and conventional TAM?
2. Figure 2B shows that the ML-DS refractory sample (L038) is present in both the erythroblast and ML-DS diagnosis clusters. However, it was not included in the erythroblast cluster at diagnosis. Do you have any speculation about the

relationship between the gene expression indicative of erythroid differentiation and refractoriness?

3. Please indicate the number of DEGs in Figure 2E and 2F as in Figure 2D.

4. The ML-DS diagnosis and relapse sample (L076) had a deletion of *CDKN2A* (loss of chromosome 9p), which is a driver mutation associated with a poor prognosis in ML-DS (see Figure 4A). The following details should be clarified:

a. Why was the loss of chromosome 9p not seen in Supplementary Figure 9?

b. Was the *CDKN2A* deletion homozygous or heterozygous?

Reviewer #3

(Remarks to the Author)

The authors investigate the risk of myeloid leukaemia in children with Down syndrome by collecting a dataset which includes transient abnormal myelopoiesis (TAM), myeloid leukaemia (ML-DS; including recurrent and refractory), and comparisons with MDS, AMKL, AML, B-ALL with/without Down syndrome. The goal of the manuscript is to identify transcriptional consequences of multi-step evolution of ML-DS through single cell sequencing and analysis. The data is interesting and the authors carefully and intentionally designed the data collection to adequately address the research questions.

My main concern from reading the paper is the lack of quantitative or statistical explanation of some of the central claims in the paper.

1. Line 179: "Nonetheless, trisomy 21 and GATA1 mutations together did not fully account for the ML-DS transcriptome." – what does it mean, to "fully account" for the transcriptome and how is this quantified?

2. Line 186: "These observations indicate that GATA1 mutations explain most of the ML-DS transcriptome, yet a final transcriptional change is required to complete the evolution of ML-DS." – what does it mean that "GATA1 mutations explain" the transcriptome and how is this quantified? In addition, does this contradict the claim that GATA1 mutations do not account for the transcriptome in line 179?

3. Line 191: "Having delineated stepwise transcriptional changes" – I do not see where the authors have provided a quantitative (or qualitative analysis) of the "steps" in transcriptional changes within the manuscript or figures. Please elaborate. Is this based on the separability of the UMAP in figure 3a, or the enrichment score in 3b/c?

Other comments:

1. Line 247: what is meant by "parallel evolution" – this calls to mind the concept of convergent evolution (compared to punctuated evolution), if this is a known concept in evolution, I would provide a reference here, and if it's meant to convey that each sample is post-relapse then I would change the terminology to avoid confusion. I'm not really sure how "parallel" evolution follows from figure 4a.

2. Line 261: same comment as above, but for "parallel lineage".

Version 1:

Reviewer comments:

Reviewer #1

(Remarks to the Author)

The authors have addressed my concerns.

(Remarks on code availability)

Reviewer #2

(Remarks to the Author)

I thank the authors for their careful revision, which adequately addressed the points raised.

(Remarks on code availability)

Reviewer #3

(Remarks to the Author)

(Remarks on code availability)

Manuscript NCOMMS-25-27940-T
Single cell transcriptional evolution of myeloid leukaemia of Down syndrome

Response to Reviewers

We would like to thank the Reviewers for their most helpful comments. We have detailed our response to each point below. Page and line numbers refer to the “clean” manuscript.

Reviewer 1

1.0	Development of myeloid leukemia in children with DS is a stepwise process that evolves from fetal liver cells with trisomy 21 to transient abnormal myelopoiesis which is defined by the acquisition of a GATA1 mutation, and finally to myeloid leukemia upon mutations in other genes such as members of the cohesion complex. This paper provides the first single cell RNA-seq analysis of this leukemia. The authors leverage many samples from the various stages of disease to define the features of this step wise leukemia development. While some of the data, such as an enrichment of megakaryocyte/erythroid lineage with decreased B-cell lineage, are confirmatory, other findings are novel and significant. Also, the comprehensive gene expression details may facilitate novel treatment strategies to prevent progression and better treat relapsed disease. The paper also provides a rich and significant dataset.	We sincerely thank the reviewer for their summary of our manuscript, as well as their thoughtful and encouraging comments.
-----	---	--

1.1	A minor limitation is that data for some of the disease subtypes are derived from a single patient. This includes comparisons of T21 with the T22 and genome wide triploid specimens as well as only one MDS sample with a GATA1 mutation (and one without), a single acute erythroid leukemia sample, and a single lymphoproliferative disease specimen. While additional samples would be helpful, these subtypes provide interesting comparisons.	We thank the reviewer for highlighting this important point. We agree that the limited sample size for certain rare disease subtypes represents a limitation in our study, and have explicitly acknowledged this in the revised manuscript (lines 308-317). Nonetheless, as the reviewer notes, these rare, unique cases provide valuable comparative insights. They enable us to assess the specificity of the transcriptional changes associated with distinct step-wise genetic lesions along the development of ML-DS - analyses that would otherwise not be feasible. Moreover, single-cell transcriptomics yields many, often thousands, of transcriptional profiles per case, which functions as within-individual biological replicates and provides high-resolution, internally consistent data that support our interpretations. That said, we fully acknowledge that single cases cannot capture the full spectrum of transcriptional heterogeneity within each condition. We hope future studies will expand on these findings with larger cohorts of these rare conditions to further validate and extend our observations. Changes to manuscript: 1) Discussion section, page 9, lines 308-317: “A key limitation of our approach is its dependence on natural variation. Accordingly, some of the comparators in our analyses are individual cases of conditions that are vanishingly rare, such as a fetus with whole genome triploidy or a disomic child with GATA1 driven MDS. While single cases certainly cannot capture the entire spectrum of transcriptional variation within each condition, they nonetheless offer critical reference points for evaluating the specificity of the transcriptional signatures associated with the genetic aberrations along the development of ML-DS. Furthermore, even in these cases, single cell transcriptomics delivered multiple readouts, which substantiates our findings at the level of individual patients. Nevertheless, larger cohorts of these rare conditions will be necessary to validate these observations and to further refine our understanding of mutation-specific transcriptional consequences.”
1.2	I'd like to see a bit more discussion of the points on lines 313-314 which mentions the possibility of using	We thank the reviewer for highlighting the importance of discussing potential biomarker candidates in more detail. The term "biomarker" in lines 313–314 primarily

some of the data as potential biomarkers. Are there any strong candidates that can be noted here?

refers to the ML-DS transcriptional signature identified in our analysis, which we propose as a composite transcriptional biomarker. This signature was found to be more highly enriched in progressive TAM compared to conventional TAM, and we think it reflects underlying leukemic programmes and neoplastic potential associated with disease progression.

While the transcriptional module as a whole may serve as a useful biomarker, we agree that identifying individual genes or gene combinations within the module with translational potential warrants further attention. Genes within the module include *CD81* (whose overexpression correlates with poor outcomes in AML¹), *CD36* (associated with metastasis and relapse in AML²), *BST2* (mediates interferon-driven HSC activation³), and *CLEC2B* (characteristically expressed in MDS-like AML blasts⁴). Such genes, particularly those encoding cell surface proteins, may be of practical clinical interest if validated at the protein level. For instance, their expression could potentially be detected using flow cytometry, facilitating minimally invasive monitoring.

A comprehensive list of the module's constituent genes is provided in **Supplementary Table 5**, with genes encoding potential cell surface markers suitable for flow cytometry indicated accordingly. However, we emphasize that further validation in larger, longitudinal cohorts is critical to account for inter-patient variability and to determine which components of the module are most robust and clinically actionable. Importantly, such validation should include correlation of transcript and protein expression levels to assess feasibility for implementation using clinically accessible assays.

Changes to manuscript:

- 1) We have added a column (“flow_cytometry_suitability”) in **Supplementary Table 5** to indicate genes encoding cell surface proteins that may be detectable using flow cytometry.
- 2) Discussion, page 10, line 340-347:

		“This indicates that the enrichment of the ML-DS transcriptional signature, including markers such as CD81, BST2, CD36, and CLEC2B, may represent potential biomarkers that enable the prospective identification of progressive TAM. Validation in larger, longitudinal cohorts will be essential to account for inter-patient heterogeneity and to refine the most robust, clinically actionable components of this transcriptional signature. While the transcriptional module as a whole may serve as a composite biomarker, further investigation is warranted to identify individual genes or gene combinations within the module that possess translational relevance.”
1.3	Also, on line 203 it would be helpful for the authors to specify that the “other myeloid leukemias” are composed of AMKL and MLL-rearranged cases (this is listed in Table 1 but would be helpful to have it mentioned in the main text).	We thank the reviewer for this suggestion. We have revised the text on line 212-213 to clarify this. Changes to manuscript: 1) Result section, page 7, lines 211-213: “Finally, we generated an orthogonal bulk transcriptomic dataset for further validation, consisting of TAM (n=16), ML-DS (n=13), other myeloid leukaemias, including AMKL (n=10) and KMT2A-rearranged leukaemia (also known as MLL-rearranged leukemia; n=28)...”
1.4	Finally, what are the karyotypes of the MDS cases with and without trisomy 21 but with and without the GATA1 mutation, and do the AMKL cases include different translocations? Is there heterogeneity among the AMKLs if they have different fusion proteins?	As noted in Figure 3A and Table 1, we have 2 MDS cases:  - L067 is a disomic child (no trisomy 21). The neoplastic cells harbour a GATA1 exon 2 mutation, which is typically associated with TAM in children with T21 (GATA1 p.T52Afs*89, Supplementary Figure 6b). This case also exhibits a copy-number neutral loss of heterozygosity on chromosome 3q, but no evidence of trisomy 21, even within the malignant compartment (Supplementary Figure 6c). - L061 is also disomic, lacking both trisomy 21 and GATA1 mutations. Instead, this case carries a somatic SETBP1 mutation (G870S), with no detectable GATA1 variants. Regarding the AMKL cases:

		In our single-cell RNA-seq dataset, we profiled two AMKL cases: - L010 harbours a KMT2A::MLLT10 fusion.- P9 harbours an RBM15::MRTFA fusion from the t(1;22) translocation, characteristic of infant AMKL. In the bulk transcriptomic dataset, out of 10 AMKL samples, two cases (3806 and 3865) harbour t(1;22) translocation resulting in RBM15::MRTFA fusion. To address the reviewer's question on heterogeneity: As shown below, at the bulk expression level, AMKL samples with and without fusion events show overlapping enrichment patterns for the ML-DS transcriptional signature. While the two RBM15::MRTFA cases (blue triangles) show lower enrichment, at least two other fusion-negative cases (black dots) exhibit similarly low scores. In the single-cell cohort, both AMKL cases - despite harboring different fusion drivers - demonstrate intermediate enrichment for the ML-DS transcriptional module (Figure 3C).
--	--	---

These observations suggest that, while fusion status defines critical biological and prognostic AMKL subtypes⁵, the ML-DS transcriptional signature derived from our study appears to capture pan-leukaemia, and pan-AMKL features that are not strictly fusion-dependent. Nonetheless, the distribution of enrichment strength observed in the AMKL bulk transcriptomic cohort may indeed reflect the heterogeneity in underlying biology of AMKL, and we agree with the reviewer that this warrants deeper investigation in future work.

As the focus of our current study is the transcriptional changes along the stepwise development of ML-DS, we included the AMKL data primarily to contextualize the ML-DS-like transcriptional signatures in other leukemias.

		Changes to manuscript: 1) We have added to Supplementary Table 1 the specific GATA1 mutation identified in each individual in our cohort (based on clinical records or our whole-genome sequencing data where available), as well as any fusion events observed in AMKL cases. The genotype of each individual is also listed in Supplementary Table 1.2) Result section, page 7, lines 246-249: “Extending this observation into the independent bulk cohort, where 4 TAM cases eventually progressed to ML-DS (termed “progressive TAM”), we observed the same pattern (Figure 3D)... Furthermore, consistent with the idea that the ML-DS module captures broader leukaemic features, we observed enrichment in both KMT2A-rearranged leukaemias and a subset of AMKL cases in the bulk cohort (though not stratified by the genetic fusion event).”
--	--	--

Reviewer 2

2.0	Trinh and his colleagues performed single-cell mRNA sequencing on fetal livers with various constitutional karyotypes, including trisomy 21, as well as on primary patient samples with myeloid neoplasms related to Down syndrome and other types of leukemia. The authors demonstrated the stepwise transcriptional progression to myeloid leukemia of Down syndrome (ML-DS), including the effects of trisomy 21 on fetal hepatic hematopoiesis, the development of transient abnormal myelopoiesis (TAM) caused by GATA1 mutations, the progression of TAM to ML-DS through additional mutations, and the development of progressive, fatal ML-DS. They showed that GATA1-induced transcriptional changes predominate throughout the disease spectrum, including relapsed/refractory disease. This technically sound work adds to our current state of knowledge regarding the pathophysiology of ML-DS. The following minor suggestions could further improve the manuscript.	We sincerely thank the reviewer for their summary of our manuscript, as well as their thoughtful and encouraging comments.
2.1	1. It is very interesting that the ML-DS transcriptional module enrichment scores for recurrent TAM and progressive TAM were intermediate between those for conventional TAM and ML-DS. The following details should be	We thank the reviewer for finding the intermediate enrichment of the ML-DS transcriptional module in recurrent and progressive TAM cases interesting. a. Below is a summary of the clinical course of the recurrent TAM patient (L156) Patient L156 presented at 3 weeks of age with a white cell count of $122 \times 10^9/L$, hepatosplenomegaly, and respiratory distress. The patient was therefore treated with

	clarified: a. Please provide a more detailed description of the clinical course of recurrent TAM patient (L156). b. Are there statistically significant differences in the median enrichment score per-sample between progressive TAM and conventional TAM?	a course of cytarabine, which led to initial clearance of blasts. At 3 months of age, the patient developed worsening pancytopenia, and an increase in blast percentage to 8% by flow cytometry. The variant allele frequency (VAF) of the GATA1 mutation was 15%. The patient was therefore treated as a recurrence of TAM with a further course of cytarabine chemotherapy, and responded well to therapy. Following this, the same GATA1-mutant clone persisted at low levels (VAF at 5%), with normal blood counts. The patient was monitored with no further treatment. Since then, the GATA1-mutant clone has cleared, and blood counts have normalised and the patient has remained well. The patient is currently 21 months old. This case demonstrates clonal persistence and re-expansion of the GATA1-mutant clone, highlighting a spectrum of outcomes in TAM. Our observation of higher enrichment of the ML-DS transcriptional signature (a pan-leukaemic signature) in neoplastic cells of this recurrent TAM case suggests that some recurrent TAM may
--	--	--

retain partial leukemic potential without fulfilling ML-DS criteria.

b. We performed a one-tailed t-test comparing ML-DS enrichment scores in bulk transcriptomes between conventional TAM (n=4) and progressive TAM (n=4) samples (**Figure 3D**). The resulting p-value was 0.055, which is just above the conventional threshold of significance of 0.05.

Purified bulk
RNA-seq dataset

ML-DS transcriptional module

While this result does not meet statistical significance ($p < 0.05$), the small p-value supports a biological trend, particularly given the small sample size. The low p-value, despite the limited sample number, suggests the difference may become statistically significant in larger cohorts.

Due to the low power of this comparison, we were initially cautious about including inferential statistics, but we agree it is helpful to report them. We now include this result in the revised manuscript and have emphasized the need for validation in future studies with larger TAM cohorts.

Changes to manuscript:

- 1) We have added a brief summary of the clinical course of the recurrent TAM patient, and elaborated on our interpretation of the higher enrichment of the ML-DS module in this case in the Result section, page 7, lines 235-241:
“Of note, we also observed imprints of this cancer transcriptome in a child with “recurrent TAM” (L156, black arrow in **Figure 3C**). This patient experienced re-expansion of the *GATA1*-mutant clone two months after initial cytarabine treatment (8% blasts by flow cytometry), and required a second

		course of cytarabine. The mutant clone then persisted at low levels (~5%) before gradually clearing over the following 19 months. From a clinical treatment perspective, this case probably represents early ML-DS rather than TAM, and the high enrichment of the ML-DS transcriptional signature suggests that the module may capture neoplastic aggressiveness or early leukemic transformation.” 2) We have added the statistical results to Result section, page 7, lines 241-246: “Extending this observation into the independent bulk cohort, where 4 TAM cases eventually progressed to ML-DS (termed “progressive TAM”), we observed the same pattern (Figure 3D). Although limited by small sample size, ML-DS enrichment scores were higher in progressive compared to conventional TAM samples, with a near-significant difference (one-tailed t-test, p = 0.055), supporting the notion that the ML-DS transcriptomes may help delineate TAM cases at higher risk of progression.” 3) We have added to the Discussion, page 10, lines 343-345: “Validation in larger, longitudinal cohorts will be essential to account for inter-patient heterogeneity and to refine the most robust, clinically actionable components of this transcriptional signature.”
2.2	2. Figure 2B shows that the ML-DS refractory sample (L038) is present in both the erythroblast and ML-DS diagnosis clusters. However, it was not included in the erythroblast cluster at diagnosis. Do you have any speculation about the relationship between the gene expression indicative of erythroid differentiation and refractoriness?	We thank the reviewer for this insightful question. The refractory ML-DS sample (L038) indeed contains a small population of neoplastic cells with a transcriptional profile closely resembling erythroblasts, which cluster alongside normal erythroid cells (Figure 2B). Notably, this erythroid-like population was absent in the diagnostic sample from the same patient. While these cells cluster together with normal erythroblasts, we are confident that these are neoplastic. These cells carry the same GATA1 mutation as the large malignant cluster of cells from L038, whereas normal erythroblasts either express wild-type GATA1 or show no detectable GATA1 transcript (Figure below). Furthermore, this subpopulation is exclusively derived from L038, while the normal erythroid cluster comprises cells from multiple donors (Supplementary Figure 3B), supporting the distinct and clonal nature of these cells. Importantly, this clustering is unlikely to be an artifact, as varying the parameters of the clustering step (such as the number of principal components, number of neighbouring points used to construct

the manifold structure, or UMAP embedding compactness) did not affect the tendency of these mutant cells to cluster with normal erythroblasts.

We think this observation may suggest that, during disease progression or under therapeutic pressure, a subset of neoplastic cells has undergone transcriptional reprogramming toward a more erythroid-like state. Cancer cells are often considered more vulnerable to chemotherapy due to their proliferative and/or other characteristic aberrancies, it is conceivable that adopting a more "normal-like" or lineage-committed phenotype could provide a selective advantage by promoting chemoresistance or immune evasion. In this case, erythroid-like differentiation may have contributed to the survival and relapse of a neoplastic subclone.

		Changes to manuscript: 1) We have added to the Result section, page 5, lines 147-155: “As is usually seen in UMAPs of single neoplastic cell transcriptomes, neoplastic cells mostly formed patient specific clusters (Supplementary Figure 3B), further indicating that TAM blasts are transcriptionally transformed. Of note, a small population of GATA1-mutant neoplastic cells from the refractory sample of patient L038 clustered with normal erythroblasts. As these blasts share the same GATA1 mutation as the main malignant population, these cells likely represent a neoplastic subclone that has adopted an erythroid-like transcriptional state. This may indicate a more lineage-committed phenotype potentially enabling a subclone to evade chemotherapy, perhaps through a less proliferative state, although further investigation is required to substantiate this hypothesis.”
2.3	3. Please indicate the number of DEGs in Figure 2E and 2F as in Figure 2D.	We thank the reviewer for this suggestion. We have updated Figure 2E and Figure 2F to include the number of DEGs accordingly. Changes to manuscript: 1) Added the number of DEGs to Figure 2E and Figure 2F.
2.4	4. The ML-DS diagnosis and relapse sample (L076) had a deletion of CDKN2A (loss of chromosome 9p), which is a driver mutation associated with a poor prognosis in ML-DS (see Figure 4A). The following details should be clarified: a. Why was the loss of chromosome 9p not seen in Supplementary Figure 9?	We thank the reviewer for raising this important point regarding the CDKN2A deletion in sample L076. a. We apologize for the lack of clarity and accuracy in Figure 4A and Supplementary Figure 9. The deletion affecting the CDKN2A locus on chromosome 9p was indeed present, but it is relatively small (~1.1Mb on chr9:21,170,000-22,340,000). In Supplementary Figure 9A, this appeared as a very subtle drop in copy number that was not explicitly annotated, which we acknowledge may have led to confusion. The figure below highlights this region on chr9p in the original genome-wide copy number

	b. Was the CDKN2A deletion homozygous or heterozygous?	plot in Supplementary Figure 9A (marked with blue box), and a zoomed-in view of chr9p showing the focal homozygous deletion around the CDKN2A locus (panel B). Panel C shows a clear drop in coverage at this segment compared to surrounding regions on chr9 within each sample. We have updated Figure 4A to explicitly note that this is a focal deletion on chr9p, and added appropriate annotation to highlight the changes in copy number state at this segment in Supplementary Figure 9A. b. The deletion of CDKN2A was homozygous, with both lost. This was determined using PURPLE, a tool for performing tumour-only copy number calling from WGS data, taking into account B-allele frequency, read depth, and tumor purity estimates. The estimated tumour purity in all four samples here ranged from ~20-33%, closely matching the estimated GATA1-mutant clone size (based on VAF). The drop in coverage of the segment was ~20-38% of baseline levels observed in the surrounding disomic regions of chromosome 9 (panel C in figure below), consistent with a clonal homozygous deletion in the dominant leukaemic population. We agree with the reviewer that CDKN2A loss is a recurrent event in ML-DS, associated with poor prognosis and therapy resistance. We have now highlighted this in the revised Results section (lines 262-264) to emphasise its clinical relevance.
--	--	--

Changes to manuscript:

- 1) Updated **Figure 4A** with more precise genetic events
- 2) Updated **Supplementary Figure 9A** to highlight chr9p21.3 homozygous deletion. Revised “chr13q loss” to “chr13q LOH”, added “chr5p LOH” where LOH stands for loss of heterozygosity.
- 3) Highlighted the significance of *CDKN2A* homozygous deletion in L076 in the Result section, page 8, lines 259-265:

“Following diagnosis with ML-DS, child L076 had two relapses after short lived remissions (months) on each occasion. Phylogenetic analyses showed that the two relapses were derived from the diagnostic clone via a common precursor but then diverged in their genetic development, accruing mutations in parallel (**Figure 4A, Supplementary Table 6**). **The major clones of all three timepoints shared identical copy number profiles and key genetic drivers (Figure 4B, Supplementary Figure 9A-B), including homozygous loss of *CDKN2A*, a lesion associated with poor prognosis in ML-DS²⁰. None**

		of the additional substitutions or indels generated plausible driver events.”
--	--	---

Reviewer 3

3.0	The authors investigate the risk of myeloid leukaemia in children with Down syndrome by collecting a dataset which includes transient abnormal myelopoiesis (TAM), myeloid leukaemia (ML-DS; including recurrent and refractory), and comparisons with MDS, AMKL, AML, B-ALL with/without Down syndrome. The goal of the manuscript is to identify transcriptional consequences of multi-step evolution of ML-DS through single cell sequencing and analysis. The data is interesting and the authors carefully and intentionally designed the data collection to adequately address the research questions.	We sincerely thank the reviewer for their summary of our manuscript, as well as their thoughtful comments.
3.1	My main concern from reading the paper is the lack of quantitative or statistical explanation of some of the central claims in the paper. 1. Line 179: “Nonetheless, trisomy 21 and GATA1 mutations together did not fully account for the ML-DS transcriptome.” – what does it mean, to “fully account” for the transcriptome and how is this quantified?	We thank the reviewer for highlighting this ambiguity. We agree that the original wording lacked clarity and have revised the text accordingly. Here, by stating that “trisomy 21 and GATA1 mutations together did not fully account for the ML-DS transcriptome”, we intended to convey that while these initiating genetic events result in large transcriptional alterations, they do not recapitulate the full transcriptional landscape observed in overt ML-DS. ML-DS represents a genetically stepwise model of leukaemogenesis. As outlined in Figure 1A, this begins with constitutional trisomy 21 (step 1), followed by acquisition of oncogenic GATA1 mutation (step 2), and finally progression to ML-DS through additional cancer causing mutations - commonly involving the cohesin complex, JAK-STAT pathway, or epigenetic regulators⁶⁻⁸ (step 3). Our aim was to characterise the transcriptional consequences associated with each of these mutational stages (or “steps”).

		TAM neoplastic cells, which carry both trisomy 21 and GATA1 mutations, exhibit different transcriptional features to ML-DS leukaemic cells. These additional differences likely reflect the effect of subsequent cooperating mutations required for full leukemic transformation. The presence of unique gene expression signatures in ML-DS, not observed in TAM, supports this conclusion (Figure 2F). We have revised the text to clarify this statement. Changes to manuscript: 1) Result section, page 6, line 187-188: “Nonetheless, trisomy 21 and GATA1 mutations together did not fully account for the ML-DS transcriptome recapitulate the gene expression profile, and thus the phenotype, of overt ML-DS.”
3.2	2. Line 186: “These observations indicate that GATA1 mutations explain most of the ML-DS transcriptome, yet a final transcriptional change is required to complete the evolution of ML-DS.” – what does it mean that “GATA1 mutations explain” the transcriptome and how is this quantified? In addition, does this contradict the claim that GATA1 mutations do not account for the transcriptome in line 179?	We apologise again for the lack of clarity in our statement. By stating that “GATA1 mutations explain most of the ML-DS transcriptome”, we refer to the observation that the transcriptional changes associated with GATA1 mutation is the most extensive among the sequential genetic alterations studied, therefore “the majority”. This is demonstrated by our analysis of differentially expressed genes (DEGs), comparing each mutational stage to the one directly preceding it (Figure 2D-F). Although these comparisons use different baselines (e.g., fetal liver MEMP cells for trisomy 21 event, trisomy 21 cells for GATA1-mutation event, and TAM cells for ML-DS), the general trend demonstrates that acquisition of GATA1 mutations results in the largest transcriptional perturbation in terms of DEG count. To better quantify this:  • ~12% of the DEGs between ML-DS and diploid fetal liver MEMP cells are already significantly altered in trisomy 21 fetal liver MEMP cells. • ~83% of the DEGs between ML-DS and trisomy 21 fetal liver MEMP cells are already present in TAM cells, consistent with GATA1 mutation driving the majority of transcriptional changes. • An additional 198 genes are differentially expressed between TAM and ML-

		DS, representing transcriptional consequences of the final leukemogenic hits. We have attempted to summarise this quantification in the diagram below, with the caveat that the transcriptional signatures induced by different mutations are not directly comparable, as the comparisons use different baselines. These findings illustrate that while GATA1 mutations drive the majority of transcriptional reprogramming along the progression towards overt ML-DS, they are insufficient to recapitulate the complete ML-DS gene expression programme. Therefore, there is no contradiction between the statements on lines 179 and 186. Rather, they describe different facets of the stepwise transcriptional progression: • Line 179 emphasizes that trisomy 21 and GATA1 mutations alone are insufficient to fully transform cells into leukaemia, and are insufficient to produce the full ML-DS phenotype at the transcriptomic level.• Line 186 highlights that GATA1 mutation is responsible for the largest transcriptional shift, thus substantially shaping the transcriptome towards that of ML-DS leukaemic cells. However, a final transformation step is needed to reach the full ML-DS phenotype. This is also reflected in the fact that TAM neoplastic cells are more similar to leukaemic phenotype than normal. We have revised both lines in the Results section to clarify these points, explicitly referring to the figures and analyses supporting these observations.
--	--	---

Asterisk (*) indicates that the reported number of differentially expressed genes between ML-DS leukaemic cells and diploid fetal liver MEMP cells may be underestimated due to the limited sample size.

Changes to manuscript:

- 1) Result section, page 6, line 194-196:

“These observations indicate that while *GATA1* mutations resulted in the most extensive transcriptional shift towards the ML-DS leukaemic state explain most of the ML-DS transcriptome, yet a final transcriptional change is required to complete the evolution of ML-DS (Figure 2D-F).”

3.3	3. Line 191: “Having delineated stepwise transcriptional changes” – I do not see where the authors have provided a quantitative (or qualitative analysis) of the “steps” in transcriptional changes within the manuscript or figures. Please elaborate. Is this based on the separability of the UMAP in figure 3a, or the enrichment score in 3b/c?	The phrasing in our original text may not have been sufficiently clear. As described above (response to 3.1), the term “stepwise” refers to transcriptional changes associated with the sequential acquisition of mutations during progression towards ML-DS. The “steps” are defined genetically, starting from acquisition of trisomy 21, followed by specific GATA1 mutation (in TAM), and additional cancer-causing mutations resulting in ML-DS (Figure 1A). This sequence of events underpins our analytical framework, where our goal is to examine how each successive genetic aberration contributes to transcriptional reprogramming. The term “stepwise” is not based on visual separability of these cell populations in the UMAP in Figure 3A. Instead, the quantification of these “stepwise” transcriptional signatures is achieved through pair-wise differential gene expression analysis between genetically and phenotypically defined cell populations, as detailed in response 3.2. The results are also presented in Figure 2D-F, where the magnitude and direction of the transcriptional changes associated with each stage are quantified. We assess the contribution of the sequential genetic events to transcriptional reprogramming towards the overt leukaemic state transcriptionally in a semi-quantitative fashion - detailed in response 3.2. We then set out to delineate the specificity of these “stepwise” transcriptional signatures by examining for their signal (i.e. enrichment) across a diverse cohort of samples, ranging from non-leukaemic fetal tissues with different karyotypic backgrounds, to a wide-range of leukaemia subtypes (Figure 3A). This reveals that whilst the trisomy 21 and GATA1-mutation signatures are specific to the respective genetic aberrations (Figure 3B), the ML-DS - defining signature represents a more generic leukaemogenic transcriptome (Figure 3C). In light of the reviewer’s comment, we have now explicitly clarified what “stepwise transcriptional changes” refer to in the revised manuscript. Changes to manuscript:
-----	---	---

		1) Result section, page 6, line 199-201: “Having delineated the stepwise transcriptional changes associated with each sequential (stepwise) genetic lesion during the progression towards ML-DS along the progression of ML-DS, we set out to assess whether these signals are specific to ML-DS or shared amongst other leukaemias.”
3.4	Other comments: 1. Line 247: what is meant by “parallel evolution” – this calls to mind the concept of convergent evolution (compared to punctuated evolution), if this is a known concept in evolution, I would provide a reference here, and if it’s meant to convey that each sample is post-relapse then I would change the terminology to avoid confusion. I’m not really sure how “parallel” evolution follows from figure 4a. 2. Line 261: same comment as above, but for “parallel lineage”.	We thank the reviewer for highlighting the ambiguity in our use of the terms “parallel evolution” and “parallel lineage”. In the context of this study, we used “parallel evolution” to describe the emergence of genetically-related but independently diverging subclones from a common ancestral precursor within the same patient. These subclones accumulate distinct (private) somatic alterations (“parallel evolution”), may persist in parallel (co-exist) and continue to expand into “parallel lineages”. This may lead to divergent trajectories in terms of their genetic, transcriptional, and phenotypic characteristics, but it is also not inconceivable that these subclones may converge on their phenotypes. These subclones may respond differently to selective pressures, such as chemotherapy where some subclones may be eliminated while others persist and expand - thus observed in relapsed disease. This usage aligns with established definitions in cancer evolution literature, where parallel evolution refers to the independent acquisition of mutations in separate subclones that may contribute to similar or different phenotypic outcomes. Unlike convergent evolution, where distinct clones independently acquire the same mutations or phenotypes, parallel evolution allows for divergence at the genomic level, regardless of whether the resulting phenotypes converge or differ. This terminology has been used to describe the behaviour across various cancers (see Turajlic et al., 2018⁹, Gatenby et al., 2025¹⁰, Coorens et al., 2020¹¹), including leukemia (e.g., Morita et al., 2020¹²). In Figure 4A, each bifurcation in the phylogenetic tree denotes a divergence event.

This results in subclones that share truncal mutations inherited from a common ancestor, but acquire private mutations that are not present in sibling branches. For example, in the box highlighted below, two subclones co-existed at first relapse of patient L076. One was eliminated by chemotherapy, while the other - harboring distinct mutations - survived and expanded, ultimately causing the second relapse. The selective expansion of this lineage is supported by the exclusive presence of SBS86, a chemotherapy-associated mutational signature, in the second relapse. This illustrates how genetically distinct subclones, originating from a shared precursor, can evolve in parallel into different lineages, but with differential outcomes in response to treatment.

The term “parallel lineage” refers to the independent, co-existing subclonal branches evolving from a common ancestor. In line 261, it refers to the diverging subclones present in the diagnostic sample of patient L038 (bifurcation in **Figure 4D**), representing sub-clonal lineages evolving independently and in parallel.

We appreciate the reviewer’s helpful feedback and have revised the manuscript to clarify these terminologies.

Changes to manuscript:

- 1) Result section, page 8, line 268-270:
 “However, despite their subsequent parallel evolution - **marked by the independent acquisition of private mutations**, they remained transcriptionally similar to each other.”
- 2) Result section, page 8, line 282-284:
 “However, this refractory clone had not been derived from the diagnostic clone but represented a parallel lineage - **a subclonal population that co-**

		existed with, and evolved independently of, the major clone present at diagnosis.”
--	--	--

Reference

1. Gonzales, F. *et al.* Tetraspanin CD81 Supports Cancer Stem Cell Function and Represents a Therapeutic Vulnerability in Acute Myeloid Leukemia. *Blood* **142**, 168 (2023).
2. Farge, T. *et al.* CD36 Drives Metastasis and Relapse in Acute Myeloid Leukemia. *Cancer Res.* **83**, 2824–2838 (2023).
3. Florez, M. A. *et al.* Interferon Gamma Mediates Hematopoietic Stem Cell Activation and Niche Relocalization through BST2. *Cell Rep.* **33**, 108530 (2020).
4. Dufva, O. *et al.* Immunogenomic Landscape of Hematological Malignancies. *Cancer Cell* **38**, 380-399.e13 (2020).
5. de Rooij, J. D. E. *et al.* Recurrent abnormalities can be used for risk group stratification in pediatric AMKL: a retrospective intergroup study. *Blood* **127**, 3424–3430 (2016).
6. Yoshida, K. *et al.* The landscape of somatic mutations in Down syndrome–related myeloid disorders. *Nat. Genet.* **45**, 1293–1299 (2013).
7. Labuhn, M. *et al.* Mechanisms of Progression of Myeloid Preleukemia to Transformed Myeloid Leukemia in Children with Down Syndrome. *Cancer Cell* **36**, 123-138.e10 (2019).
8. Sato, T. *et al.* Landscape of driver mutations and their clinical effects on Down syndrome–related myeloid neoplasms. *Blood* **143**, 2627–2643 (2024).
9. Turajlic, S. *et al.* Deterministic Evolutionary Trajectories Influence Primary Tumor Growth: TRACERx Renal. *Cell* **173**, 595-610.e11 (2018).
10. Gatenby, R. A., Teer, J. K., Tsai, K. Y. & Brown, J. S. Parallel and convergent dynamics in the evolution of primary breast and lung adenocarcinomas. *Commun. Biol.* **8**, 775 (2025).
11. Coorens, T. H. H. *et al.* Lineage-Independent Tumors in Bilateral Neuroblastoma. *N. Engl. J. Med.* **383**, 1860–1865 (2020).

12. Morita, K. *et al.* Clonal evolution of acute myeloid leukemia revealed by high-throughput single-cell genomics. *Nat. Commun.* **11**, 5327 (2020).

Single cell transcriptional evolution of myeloid leukaemia of Down syndrome

Response to Reviewers Comments

Reviewer #1 (Remarks to the Author): The authors have addressed my concerns.	We thank the reviewer for their positive assessment and are pleased that our revisions have addressed all concerns.
Reviewer #2 (Remarks to the Author): I thank the authors for their careful revision, which adequately addressed the points raised.	We thank the reviewer for the constructive feedback and are pleased that our revisions have addressed all of the points raised.